# ProRL: Prolonged Reinforcement Learning Expands Reasoning Boundaries in Large Language Models

**Mingjie Liu**    **Shizhe Diao**    **Ximing Lu**    **Jian Hu**    **Xin Dong**
**Yejin Choi**    **Jan Kautz**    **Yi Dong**
NVIDIA
{mingjiel, sdiao, ximingl, jianh, xind, yejinc, jkautz, yidong}@nvidia.com

## Abstract

Recent advances in reasoning-centric language models have highlighted reinforcement learning (RL) as a promising method for aligning models with verifiable rewards. However, it remains contentious whether RL truly expands a model's reasoning capabilities or merely amplifies high-reward outputs already latent in the base model's distribution, and whether continually scaling up RL compute reliably leads to improved reasoning performance. In this work, we challenge prevailing assumptions by demonstrating that prolonged RL (**ProRL**) training can uncover novel reasoning strategies that are inaccessible to base models, even under extensive sampling. We introduce ProRL, a novel training methodology that incorporates KL divergence control, reference policy resetting, and a diverse suite of tasks. Our empirical analysis reveals that RL-trained models consistently outperform base models across a wide range of pass@$k$ evaluations, including scenarios where base models fail entirely regardless of the number of attempts. We further show that reasoning boundary improvements correlates strongly with task competence of base model and training duration, suggesting that RL can explore and populate new regions of solution space over time. These findings offer new insights into the conditions under which RL meaningfully expands reasoning boundaries in language models and establish a foundation for future work on long-horizon RL for reasoning. We release model weights to support further research:

https://huggingface.co/nvidia/Nemotron-Research-Reasoning-Qwen-1.5B

## 1   Introduction

Recent advances in reasoning-focused language models, exemplified by OpenAI-O1 [1] and DeepSeek-R1 [2], have marked a paradigm shift in artificial intelligence by scaling test-time computation. Specifically, test-time scaling enables long-form Chain-of-Thought (CoT) thinking and induces sophisticated reasoning behaviors, leading to remarkable improvements on complex tasks such as mathematical problem solving [3–6] and code generation [7, 8]. By continuously expending compute throughout the reasoning process—via exploration, verification, and backtracking—models boost their performance at the cost of generating longer reasoning traces.

At the heart of these advances lies reinforcement learning (RL), which has become instrumental in developing sophisticated reasoning capabilities. By optimizing against verifiable objective rewards rather than learned reward models, RL-based systems can mitigate the pitfalls of reward hacking [9–11] and align more closely with correct reasoning processes. However, a fundamental question remains under active debate within the research community: *Does reinforcement learning truly*

39th Conference on Neural Information Processing Systems (NeurIPS 2025).

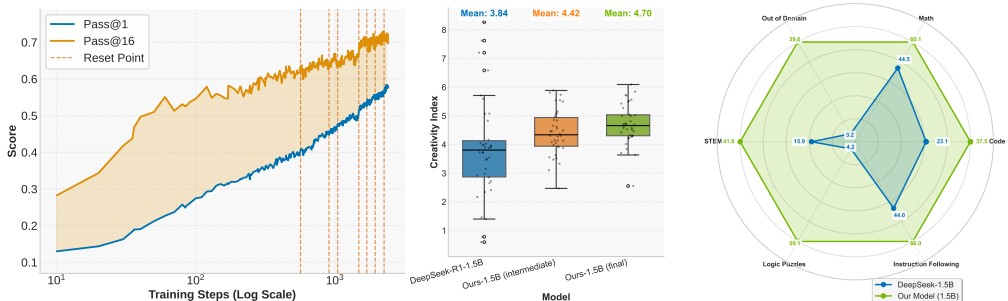

Figure 1: Benefits of prolonged reinforcement learning (ProRL). **Left**: Pass@1 and Pass@16 scales with ProRL training. **Middle**: ProRL leads to more novel solutions reflected by higher Creativity Index [12]. **Right**: Our model greatly surpass base model across diverse tasks.

*unlock new reasoning capabilities from a base model, or does it merely optimize the sampling efficiency of solutions already embedded in the base model?*

Recent studies [13–15] argues for the latter, claiming that RL-trained models do not acquire new reasoning capabilities beyond what exists in their base models based on pass@$k$ metrics. We posit that these conclusions may stem from methodological constraints rather than fundamental limitations of RL approaches themselves. Specifically, we identify two key limitations in existing research: (1) an overreliance on specialized domains like mathematics, where models are often overtrained during both pre-training and post-training phases, thereby restricting the potential for exploration; and (2) the premature termination of RL training before models can fully explore and develop new reasoning capabilities based on a limited amount of RL training, typically no more than hundreds of steps [13].

In this study, we address these limitations through several key contributions. First, we introduce ProRL, a recipe designed to enable extended RL training periods that facilitate deeper exploration of reasoning strategies. It enables more than 2k training steps and scale the training data across diverse tasks—from traditional math and code tasks to STEM problems, logical puzzles, and instruction following, which, we hypothesize, are crucial for generalization. Using ProRL, we developed **Nemotron-Research-Reasoning-Qwen-1.5B**, **the world's best 1.5B reasoning model** that significantly outperforms its base model, DeepSeek-R1-1.5B, and matches or even surpasses the performance of DeepSeek-R1-7B across a diverse range of benchmarks. Notably, compared to DeepSeek-R1-1.5B, we achieve average pass@1 improvements of 14.7% on math benchmarks, 13.9% on coding, 54.8% on logic puzzles, 25.1% on STEM reasoning, and 18.1% on instruction-following tasks (Figure 1, Right). More importantly, ProRL demonstrates continued performance improvements after an unprecedented 2k training steps (Figure 1, Left), suggesting that *RL training scales effectively with increased compute*.

Furthermore, Nemotron-Research-Reasoning-Qwen-1.5B offers surprising new insights —*RL can indeed discover genuinely new solution pathways entirely absent in base models, when given sufficient training time and applied to novel reasoning tasks.* Through comprehensive analysis, we show that our model generates novel insights and performs exceptionally well on tasks with increasingly difficult and out-of-domain tasks, suggesting a genuine expansion of reasoning capabilities beyond its initial training. Most strikingly, we identify many tasks where the base model fails to produce any correct solutions regardless of the amount of sampling, while our RL-trained model achieves 100% pass rates (Figure 4). Interestingly, we find the amount of gain from RL on each task is predictable given the base model's performance—RL expands a model's reasoning boundary most effectively in domains where the base model initially struggles. Moreover, we quantify the novelty of the model's reasoning trajectories using the Creativity Index [12], which measures the amount of overlap with a pretraining corpus. We find that prolonged RL training leads to trajectories with higher novelty (Figure 1, Middle), indicating the emergence of new reasoning patterns during RL.

Our findings hold significant implications for the broader AI community, demonstrating that RL approaches can indeed enhance model capabilities without requiring additional training data. Through sustained exploration, models can develop new knowledge and reasoning strategies that potentially exceed human insights. This work reaffirms the value of reinforcement learning as a pathway toward more capable and generalizable AI systems, challenging previous assumptions about the inherent limitations of these approaches.

# 2 ProRL: Prolonged Reinforcement Learning

We begin with a brief overview of the GRPO [16] algorithm. We then address key challenges in prolonged RL training, such as entropy collapse and instability, by introducing a KL divergence penalty and periodic resets of the reference policy. This ensures stable training across many epochs and continued performance improvement.

## 2.1 Background: Group Relative Policy Optimization

We adopt Group Relative Policy Optimization (GRPO) [16] as the core RL algorithm. Compared with Proximal Policy Optimization (PPO) [17], it removes the value model and instead use baseline estimates based on group scores. Formally the GRPO maximizes the following objective:

$$\mathcal{L}_{\text{GRPO}}(\theta) = \mathbb{E}_{\tau \sim \pi_\theta} \left[ \min \left( r_\theta(\tau) A(\tau), \quad \text{clip}(r_\theta(\tau), 1 - \epsilon, 1 + \epsilon) A(\tau) \right) \right], \tag{1}$$

where $\tau$ is the response sampled from the current policy $\pi_\theta$. $r_\theta(\tau) = \frac{\pi_\theta(\tau)}{\pi_{old}(\tau)}$ is the probability ratio between the current policy and old policy before each actor update. The advantage used in GRPO foregoese the critic model of PPO, and instead estimates baseline from group scores $\{R_i\}_{i \in G(\tau)}$:

$$A(\tau) = \frac{R_\tau - mean(\{R_i\}_{i \in G(\tau)})}{std(\{R_i\}_{i \in G(\tau)})}. \tag{2}$$

## 2.2 Prolonged Reinforcement Learning (ProRL)

### 2.2.1 Mitigating Entropy Collapse

A key challenge in prolonged policy optimization is entropy collapse, a phenomenon where the model's output distribution becomes overly peaked early in training, resulting in sharply reduced entropy. When entropy collapses, the policy prematurely commits to a narrow set of outputs, severely limiting exploration. This is particularly detrimental in methods like GRPO, where the learning signal depends on having a diverse set of sampled outputs to effectively estimate relative advantages. Without sufficient exploration, policy updates become biased, leading to stagnation in training.

A common mitigation strategy is to increase the sampling temperature during rollouts. However, we find that this approach only delays the onset of entropy collapse rather than preventing it altogether, as entropy continues to decline steadily as training progresses. Nonethenless, we did employ high rollout temperature since encourages exploration by increasing the initial entropy.

## 2.3 Decoupled Clip and Dynamic Sampling Policy Optimization (DAPO)

To address entropy collapse, we adopt several components from the DAPO algorithm [4], which are specifically designed to maintain exploration and output diversity. First, DAPO introduces decoupled clipping, where the lower and upper clipping bounds in the PPO objective are treated as separate hyperparameters:

$$\text{clip}(r_\theta(\tau), 1 - \epsilon_{low}, 1 + \epsilon_{high}). \tag{3}$$

By setting a higher value for $\epsilon_{high}$, the algorithm promotes 'clip-higher', uplifting the probabilities of previously unlikely tokens and encouraging broader exploration. We find that this modification helps retain entropy and reduces premature mode collapse.

Additionally, DAPO employs dynamic sampling, filtering out prompts for which the model consistently succeeds or fails (i.e., accuracy 1 or 0), as these provide no learning signal. This focus on intermediate difficulty examples further helps maintain a diverse learning signal during training.

### 2.3.1 KL Regularization and Reference Policy Reset

While DAPO and temperature adjustment help slow entropy collapse, we find that explicit regularization via a KL divergence penalty provides a stronger and more stable solution. Specifically, we incorporate a KL penalty between the current policy $\pi_\theta$ and a reference policy $\pi_{ref}$:

$$L_{KL-RL}(\theta) = L_{GRPO}(\theta) - \beta D_{KL}(\pi_\theta || \pi_{ref}). \tag{4}$$

This penalty not only helps maintain entropy but also serves as a regularizer to prevent the online policy from drifting too far from a stable reference, stabilizing learning and mitigating overfitting to spurious reward signals.

Recent works [4, 7, 5, 18] have argued for the removal of the KL penalty, citing that models naturally diverge during training on chain-of-thought reasoning tasks. We observe that this perspective often applies to base models prior to any supervised fine-tuning. In contrast, we begin from a well-initialized checkpoint (DeepSeek-R1-Distill-Qwen-1.5B) already capable of generating coherent CoT outputs. In this context, retaining a KL penalty is still beneficial for both stability and sustained entropy.

We further observe that as training progresses, the KL term may increasingly dominate the loss, leading to diminishing policy updates. To alleviate this, we introduce a simple yet effective technique: *reference policy reset*. Periodically, we hard-reset the reference policy $\pi_{ref}$ to a more recent snapshot of the online policy $\pi_\theta$, and reinitialize the optimizer states. This allows the model to continue improving while maintaining the benefits of KL regularization. We apply this reset strategy throughout training to avoid premature convergence and encourage prolonged training.

# 3 Nemotron-Research-Reasoning-Qwen-1.5B: The World's Best 1.5B Reasoning Model

We present Nemotron-Research-Reasoning-Qwen-1.5B, a generalist model trained via reinforcement learning on a diverse, verifiable dataset of 136K problems across math, code, STEM, logic puzzles, and instruction following. Leveraging stable reward computation, improved GRPO, and prolonged training, our model achieves strong generalization across domains. It outperforms DeepSeek-R1-Distill-Qwen-1.5B by +15.7% on math, +14.4% on code, +25.9% on STEM, +22.0% on instruction following, and +54.8% on text-based logic puzzles, Reasoning Gym [19]. It also surpasses domain-specialized baselines in both math (+4.6%) and code (+6.5%), demonstrating the effectiveness of generalist prolonged RL training.

## 3.1 Training Dataset

We construct a diverse and verifiable training dataset spanning 136K examples in five task domains, math, code, STEM, logical puzzles, and instruction following, to enable robust reinforcement learning from a wide range of reasoning problems. Each task type is paired with a clear reward signal (binary or continuous), allowing for reliable feedback during training. This broad task coverage encourages generalization beyond narrow domains and enables meaningful comparison of RL algorithms across diverse reward structures. Details on the composition of training dataset is presented in Appendix E.

## 3.2 Training Setup

We use verl [20] for reinforcement learning training. We adopt enhancements of GRPO [16] proposed by DAPO [4], decoupling clipping hyperparameters with $\epsilon_{low} = 0.2, \epsilon_{high} = 0.4$, and dynamic sampling for filtering prompts that are too easy or difficult (with accuracy equal to 1 and 0). For rollout, we sample $n = 16$ responses for each prompt with a context window limit of 8096 and use a high sampling temperature of 1.2. We set batch size to 256 and mini-batch size to 64 (equating to 4 gradient updates per rollout step). For training we use the AdamW [21] optimizer with a constant learning rate of $2 \times 10^{-6}$. We conduct training on 4 8 x NVIDIA-H100-80GB nodes, and the whole training runs for approximately 16k GPUs hours.

## 3.3 ProRL Training Dynamics

To enable effective long-horizon reinforcement learning, we monitor training progress using a blended validation set derived from the evaluation benchmark. When validation performance stagnates or degrades, we perform a hard reset of the reference model and optimizer. This not only restores training stability but also facilitates greater policy divergence from the base model. Throughout most of training, we cap response length at 8k tokens to maintain concise and stable generations. In the final stage (~200 steps), we increase the context window to 16k tokens, observing that the model adapts quickly and achieves measurable improvements. We detail our training recipe in Appendix F.

Table 1: Performance (pass@1) comparison for benchmarks across Math domain. The best results are highlighted in **bold**. The results of DeepSeek-R1-Distill-Qwen-7B are marked as gray and are provided as a reference (same in all following tables).

| Model | AIME24 | AIME25 | AMC | Math | Minerva | Olympiad | Avg |
|---|---|---|---|---|---|---|---|
| DeepSeek-R1-Distill-Qwen-1.5B | 28.54 | 22.71 | 62.58 | 82.90 | 26.38 | 43.58 | 44.45 |
| DeepScaleR-1.5B | 40.21 | 31.46 | 73.04 | 89.36 | 41.57 | 51.63 | 54.54 |
| DeepSeek-R1-Distill-Qwen-7B | 53.54 | 40.83 | 82.83 | 93.68 | 50.60 | 57.66 | 63.19 |
| Nemotron-Research-Reasoning-Qwen-1.5B | **48.13** | **33.33** | **79.29** | **91.89** | **47.98** | **60.22** | **60.14** |

Table 2: Performance (pass@1) comparison across benchmarks for Code. We abbreviate benchmarks names for codecontests (cc), codeforces (cf), humanevalplus (human), and livecodebench (LCB).

| Model | apps | cc | cf | taco | human | LCB | Avg |
|---|---|---|---|---|---|---|---|
| DeepSeek-R1-Distill-Qwen-1.5B | 20.95 | 16.79 | 14.13 | 8.03 | 61.77 | 16.80 | 23.08 |
| DeepCoder-1.5B | 30.37 | 23.76 | 21.70 | 13.76 | **73.40** | 22.76 | 30.96 |
| DeepSeek-R1-Distill-Qwen-7B | 42.08 | 32.76 | 33.08 | 19.08 | 83.32 | 38.04 | 41.39 |
| Nemotron-Research-Reasoning-Qwen-1.5B | **41.99** | **31.80** | **34.50** | **20.81** | 72.05 | **23.81** | **37.49** |

Table 3: Performance comparison on STEM reasoning (GPQA Diamond), instruction following (IFEval), and logic puzzles (Reasoning Gym) tasks. We also present results on OOD tasks: *acre*, *boxnet*, and *game_of_life_halting* (game).

| Model | GPQA | IFEval | Reasoning | acre | boxnet | game |
|---|---|---|---|---|---|---|
| DeepSeek-R1-Distill-Qwen-1.5B | 15.86 | 44.05 | 4.24 | 5.99 | 0.00 | 3.49 |
| DeepSeek-R1-Distill-Qwen-7B | 35.44 | 58.01 | 28.55 | 20.21 | 1.71 | 12.94 |
| Nemotron-Research-Reasoning-Qwen-1.5B | **41.78** | **66.02** | **59.06** | **58.57** | **7.91** | **52.29** |

Figure 2 illustrates key statistics on training dynamics over the course of extended reinforcement learning across multiple stages. The application of various enhancements proposed by DAPO [4], along with the inclusion of KL divergence loss, helped the model avoid entropy collapse. Although we observe a positive correlation between average response length and validation scores, this factor does not appear to be decisive, as there are training stages where performance improves without requiring longer responses. Meanwhile, the validation performance, measured by both pass@1 and pass@16, consistently improved and scaled with increased training computation.

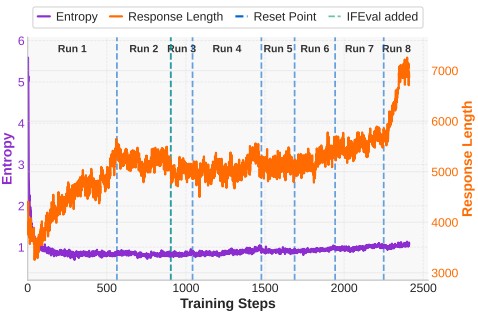

Figure 2: ProRL training dynamics.

## 3.4 Evaluation

**Evaluation Benchmarks.** We evaluate models on the breadth of various tasks across math, coding, reasoning, and instruction following. For math, we follow DeepScaleR [3] and SimpleRL [22], and evaluate on AIME2024 [23], AIME2025 [24], AMC [25] (composed of AMC2022 and AMC2023), MATH [26], Minerva Math [27], and Olympiad Bench [28]. For coding, we use the validation set from PRIME [29] consisted of APPS [30], Codecontests [31], Codeforces[1], and TACO [32]. We also include benchmarks HumanevalPlus [33] and LiveCodeBench [34]. For logic puzzles, we reserved 100 samples from each reasoning gym [19] tasks as test datasets for evaluation. In addition, we use a curated subset[2] from GPQA Diamond [35] and IFEval [36] to evaluate the capability of our models in STEM reasoning and instruction following [37].

---

[1]https://huggingface.co/datasets/MatrixStudio/Codeforces-Python-Submissions
[2]https://huggingface.co/datasets/spawn99/GPQA-diamond-ClaudeR1

**Evaluation Settings.** We use vllm [38] as the inference backend, with a sampling temperature of 0.6, nucleus sampling [39] with $top\_p = 0.95$ and maximum response length of 32k. For math, coding, and STEM reasoning tasks, we obtain estimates of pass@1 from 16 samples for each benchmark prompt from strictly binary rewards. For other tasks (logical puzzles and instruction following), we calculate the average continuous reward score from our rule-based verifiers. We evaluate and report benchmark results for open-source models using our own evaluation settings.

**Evaluation Results.** We provide a detailed comparison between DeepSeek-R1-Distill-Qwen-1.5B and our final model Nemotron-Research-Reasoning-Qwen-1.5B across multiple domains. In the math domain shown in Table 1, our model consistently outperforms the base model across benchmarks, showing an average improvement of 15.7%. For code domain results shown in Table 2, our final model surpasses the base model in competitive programming tasks as measured by $pass@1$ accuracy by 14.4%. Our model also demonstrates substantial gains in STEM reasoning and instruction following, with improvements of 25.9% on GPQA Diamond and 22.0% on IFEval. Our model achieves high accuracy on Reasoning Gym logic puzzles after training, despite the base model struggles with formatting and challenging subtasks, improving reward by 54.8%. Even compared to a much larger model, DeepSeek-R1-Distill-Qwen-7B, our model achieves comparable or even better performance across multiple domains.

**Generalization to OOD Tasks.** In Table 3, we also present results on out-of-distribution (OOD) tasks in Reasoning Gym. Our model shows significant improvements on three OOD tasks, demonstrating stronger generalization beyond the training distribution. This highlights the effectiveness of our training approach in enabling the model to adapt and perform well on unseen challenges.

**Comparision with Domain-Specialized Models.** We compare the performance of Nemotron-Research-Reasoning-Qwen-1.5B with two domain-specialized baselines: DeepScaleR-1.5B [3], tailored for mathematical reasoning, and DeepCoder-1.5B [7], focused on competitive programming tasks. Our ProRL trained model enables strong generalization, achieving superior pass@1 scores on both math (+4.6%) and code (+6.5%) benchmarks. Additionally, ProRL enables deeper exploration and refinement within limited response length, where prior works often increase training response length too early, causing "overthinking" [40] with verbose reasoning.

## 4 Analysis: Does ProRL Elicit New Reasoning Patterns?

To evaluate whether prolonged ProRL training enhances reasoning beyond the base model, we increase inference samples to 256 and re-evaluate performance. Due to compute limits, we randomly select 18 Reasoning Gym tasks (out of 96) and re-run all other benchmarks: math, code, STEM reasoning, and instruction following. We compare the base model (DeepSeek-R1-Distilled-1.5B), an intermediate checkpoint, and Nemotron-Research-Reasoning-Qwen-1.5B (the final model after extended training).

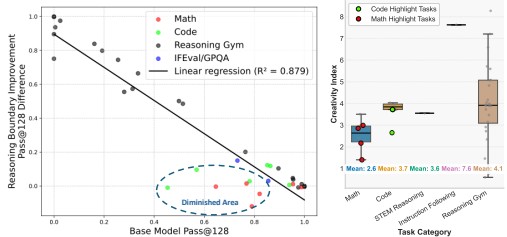

Figure 3: **Left**: ProRL expands a model's reasoning boundary most effectively on tasks where the base model initially struggles. **Right**: Tasks with minimal gains post-RL highlighted in the circle tend to have a lower creativity index, indicating higher overlap with pretraining data.

### 4.1 The Weaker the Start, the Stronger the Gain with ProRL

A key finding from our study is that the effectiveness of RL in expanding a model's reasoning boundary (measured by pass@128) is strongly influenced by the base model's initial capabilities. As shown in Figure 3, we observe a significant negative correlation between the base model's reasoning boundary and the extent of reasoning improvement after RL training. Specifically, tasks where the base model already performs well (i.e., high pass@128) tend to exhibit minimal or even negative gains in reasoning breadth post-RL. This indicates a narrowing of the reasoning boundary, where the model becomes more confident in a subset of solutions it already understands, rather than exploring new reasoning patterns. In contrast, in domains where the base model struggles, particularly those with a low initial pass@128, RL training is most effective. Here, ProRL not only improves pass@1,

but also expands the model's ability to explore and succeed in a broader range of reasoning paths. To further confirm our intuition that tasks with minimal gains post-RL are those the base model is familiar with, we compute the creativity index [41] of the base model's responses for each task against the largest open-source pretraining corpus, DOLMA [42]. The creativity index quantifies the degree of overlap between model's responses and the some math and code tasks highlighted in the circle—tend to have lower creativity indices, suggesting the base model has seen a large amount of similar data during pretraining.

## 4.2 Unpacking ProRL's Reasoning Boundaries: Diminish, Plateau, and Sustained Gains

We analyze performance trends on individual benchmarks and categorize them based on how pass@$k$ evolves throughout training. Our analysis reveals that reinforcement learning can meaningfully expand a model's reasoning capacity, particularly on challenging tasks that extend beyond the capability of the base model. While some tasks exhibit early saturation or even regressions in reasoning breadth, we also observe clear instances where the model's reasoning capabilities expand with continued training. Most notably, on some domains such as code generation, ProRL enables continued gains, suggesting that prolonged training allows the model to explore and internalize more sophisticated reasoning patterns. This demonstrates that, under the right conditions, ProRL can push the frontier of a model's reasoning abilities beyond what the base model achieves.

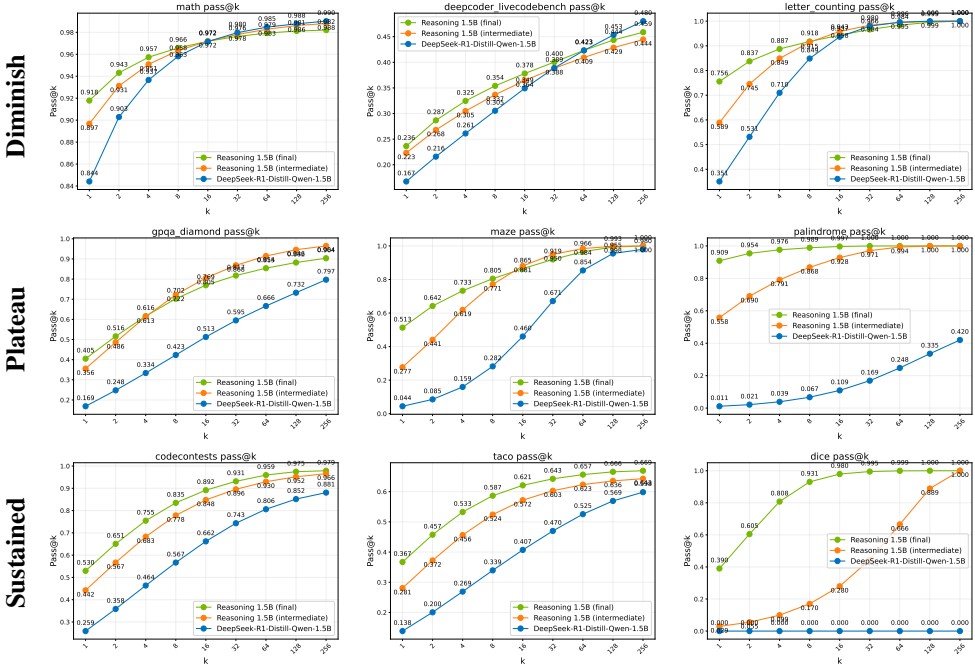

Figure 4: Pass@$k$ comparison of the base model, an intermediate checkpoint, and the final RL-trained models. Trends are grouped into three regimes: (1) **Diminish**: reduced diversity due to narrow output distributions; (2) **Plateau**: early RL saturation of gains in reasoning boundary; and (3) **Sustained**: continued reasoning boundary improvement with prolonged training.

**Diminished Reasoning Boundary** In some benchmarks (particularly in the math domain), Nemotron-Research-Reasoning-Qwen-1.5B exhibit decreased or unchanged reasoning capacity compared to the base model, aligning with observations of prior work [13]. Although pass@1 improves, the pass@128 score, which reflects broader reasoning ability, often declines. These tasks tend to have a high baseline pass@128, suggesting that the base model already possesses sufficient reasoning ability, and RL training merely sharpens the output distribution at the expense of exploration and generality.

**Gains Plateau with RL** For these tasks, RL training boosts both pass@1 and pass@128, indicating improved reasoning. However, these gains are largely achieved early in training. Comparing the intermediate and final checkpoints shows that ProRL offers negligible additional benefit, implying that the model quickly saturates its learning potential for these tasks.

**Sustained Gains from ProRL** In contrast, some benchmarks, particularly more complex ones such as coding, Nemotron-Research-Reasoning-Qwen-1.5B show continued improvements in reasoning capacity with prolonged RL training. These tasks likely require extensive exploration of diverse problem instances during training to generalize effectively to the test set. In such cases, ProRL expands the model's reasoning boundaries.

### 4.3 ProRL Enhances Out-of-Distribution Reasoning

We focus on how ProRL influences the model's ability to generalize beyond the distribution of its training data. These studies aim to isolate the role of extended RL updates in expanding the model's reasoning boundaries, especially on structurally novel or semantically challenging tasks that were not encountered during initial training.

**Out-of-Distribution (OOD) Task** We evaluate the model on Reasoning Gym task *boxnet*, which was not seen during training. As shown in Figure 5 (Check Appendix D.3 for an example), the base model exhibits no capability of solving the task. In contrast, the model trained with ProRL demonstrates a significant ability to solve the problem, indicating a clear expansion in the model's reasoning boundary, generalizing to out-of-distribution tasks unseen during training. Furthermore, when comparing an intermediate RL checkpoint with the final prolonged RL model, we observe that extended training sustains and amplifies performance gains consistently across all values of $k$. These results further support the conclusion that ProRL enables the model to internalize abstract reasoning patterns that generalize beyond specific training distributions or complexity levels.

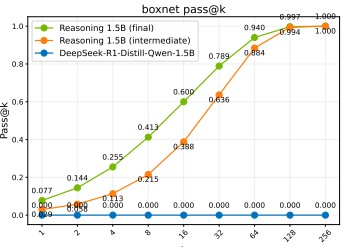

Figure 5: Expanded reasoning boundary for OOD task *boxnet*.

**Increased Task Difficulty** We evaluate performance across varying levels of task difficulty for *graph_color* task (Check Appendix D.1 for an example) by generating graph problems with different numbers of graph nodes. While the training data only includes graphs of size 10, we test on larger graphs to assess generalization beyond the training regime. Figure 6 plots the pass@1 (solid lines) and pass@128 (dashed lines) across different models. The results reveal a consistent decline in performance as task difficulty increases, which is expected given the combinatorial growth in solution space. However, our prolonged RL model maintains significantly higher accuracy across all graph sizes compared to both the base and intermediate models. This indicates that extended RL updates not only enhance pass@1 on

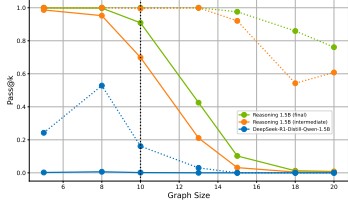

Figure 6: ProRL generalizes to increased task difficulty on *graph_color*.

in-distribution tasks but also improve the model's robustness to more complex, unseen scenarios.

### 4.4 How Does pass@1 Distributions Evolve as ProRL Progresses?

Dang et al [14] derived a mathematical upper bound for pass@$k$ as:

$$\mathbb{E}_{x,y\sim D}[pass@k] \leq 1 - \left((1 - \mathbb{E}_{x,y\sim D}[\rho_x])^2 + \text{Var}(\rho_x)\right)^{k/2}, \tag{5}$$

where $\rho_x$ represents the pass@1 accuracy for task $x$. While increasing expected pass@1 raises this upper bound, higher variance reduces it. In contrast to [14]'s observation of declining pass@$k$ during training, our results in Figure 1 demonstrate continuous improvement in both pass@1 and pass@16, reproducing the scaling law patterns reported for OpenAI O1's RL training [43]. Our ProRL approach generates substantial performance gains across diverse tasks. Figures 7(a) and 7(b) illustrate significant rightward distribution shifts in code and logic puzzle tasks. Initially concentrated near zero with extended tails, the pass@1 distributions evolved markedly after training. Codeforces problems exhibit broader distribution patterns post-training, while the *family_relationships* task (Appendix D.2 for an example), which is a novel reasoning challenge, demonstrate a dramatic shift from predominantly zero accuracy to peaking at perfect accuracy, indicating successful solution discovery across the majority of prompts. These pronounced distribution changes, driven by extended RL training, produce sufficient improvement in expected pass@1 to overcome any negative effects from increased variance.

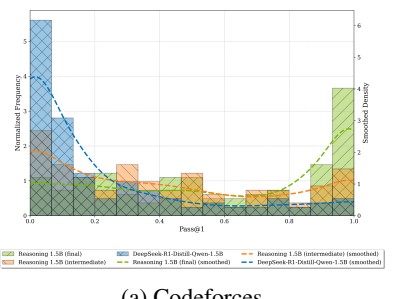

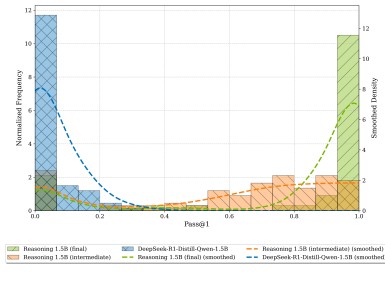

| (a) Codeforces | (b) *family_relationships* |

Figure 7: Distribution shifts in pass@1 accuracy following prolonged RL training across two representative tasks. The figure illustrates the evolution of pass@1 probability distributions for selected tasks from code (a) codeforces, and reasoning domains (b) *family_relationships*.

## 5 Related Work

**Reasoning Models** Reasoning models represent a specialized category of AI systems that engage in detailed, long chain-of-thought before generating final answers, a concept first introduced by OpenAI's o1 series models [44]. Subsequently, DeepSeek [2] and Kimi [45] detail methodologies for training reasoning models using reinforcement learning with verifiable rewards (RLVR). Both approaches have popularized RL algorithms like GRPO [16], Mirror Descent[46], RLOO [47] and other variants. While numerous open-source efforts have attempted to reproduce o1-like models, most focus on single domains [3, 7, 6] or study test-time compute scaling [48], with few addressing prolonged reinforcement learning training or examining RL training time scaling laws. As widely acknowledged in the reinforcement learning community, RL training presents significant challenges due to its sensitivity to hyperparameters [49]. Various reinforcement learning techniques [5, 4] have been studied to enhance training stability for sustained optimization periods. Our research demonstrates that achieving prolonged RL training can substantially expand the boundaries of reasoning capabilities in these models.

**RL Reasoning Boundary** Achieving superhuman performance has been the holy grail of machine learning, with reinforcement learning algorithms successfully delivering on this expectation, starting with DeepQ networks for Atari games [50, 51]. More recently, AlphaGo and AlphaZero [52] have demonstrated that AI agents can enhance their performance indefinitely by continuously iterating between data collection via Monte Carlo Tree Search and policy improvement. These examples show that RL training helps agents develop novel techniques not present in their base models [53–57]. However, challenging this perspective, several recent studies question whether RL training genuinely enhances the reasoning capacity of LLMs. One work [13] argue that the RLVR method fails to extend this capacity, as evidenced by pass@$k$ metrics showing no improvement and in some cases deterioration, compared to the base model, a trend echoed by other researchers [14]. Similarly, another work [15] finds that RL algorithms tend to converge toward a dominant output distribution, merely amplifying existing pretraining patterns. Beyond pass@$k$ metrics, alternative measurements like creativity index [12] can also determine whether models learn new ideas through RL training, which we employ during our studies.

## 6 Conclusion

In this work, we address whether reinforcement learning can truly expand language models' reasoning boundaries. Through our introduction of ProRL, we provide compelling evidence that extended, stable RL training develops novel reasoning patterns beyond a base model's initial capabilities. ProRL incorporates KL divergence penalties and periodic reference policy resets to maintain training stability over long durations. Using this approach, we developed a state-of-the-art 1.5B parameter generalist reasoning model trained on diverse datasets spanning mathematics, coding, STEM, logical puzzles, and instruction following tasks. Our analysis reveals ProRL is particularly effective for tasks where the base model initially struggles. Most importantly, ProRL enables strong generalization to out-of-distribution tasks and increasingly complex problems, demonstrating that extended RL training helps models internalize abstract reasoning patterns transferable beyond the training distribution.

These results challenge previous assumptions about RL's limitations and establish that sufficient training time with appropriate techniques can meaningfully expand reasoning boundaries, providing valuable direction for development of more capable reasoning models.

## Acknowledgments

We sincerely thank Shrimai Prabhumoye for the insightful discussions that significantly inspired our work. We are also grateful to Sahil Jain for sharing his perspectives on addressing entropy collapse. Additionally, we thank Makesh Narsimhan Sreedhar and David Mosallanezhad for their assistance with running several model evaluations.

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

## A Detailed Ablation Studies

We refer readers to our technical report [58] for detailed ablation studies, including analyses of hyperparameters and policy resets. Additional results on training with different base models and scaling to larger models and compute are provided in our updated blog post [59].

## B Limitations

Despite the impressive results achieved by our ProRL approach, several important limitations should be acknowledged:

**Computational Resources**    The extended RL training process requires substantial computational resources, which may be prohibitive for smaller organizations or researchers with limited budgets. Our approach involves multiple training stages with periodic resets, long reasoning chains sampling further intensifying these requirements.

**Scalability Concerns**    While we demonstrate effective training of a 1.5B parameter model, it remains unclear how well our approach scales to larger models. The increase in computational requirements becomes more pronounced with larger parameter counts.

**Training Process Challenges**    Our approach requires periodic hard-resets of the reference policy and optimizer parameters to maintain training stability. This introduces additional complexity to the training process and may lead to inconsistent results compared to more stable training methods.

**Limited Task Scope**    While our evaluation covers diverse domains, the training dataset still represents only a subset of possible reasoning tasks. The performance on certain out-of-distribution tasks shows promising generalization, but we cannot guarantee similar improvements across all potential reasoning domains not explicitly included in our training or evaluation.

## C Societal Impacts

The development of Prolonged Reinforcement Learning (ProRL) has significant implications for both the AI research community and society at large. By enhancing reasoning capabilities of language models across domains, this approach creates both opportunities and challenges that warrant careful consideration.

**Potential Benefits and Opportunities**    ProRL demonstrates that current RL methodology can potentaily achieve superhuman reasoning capabilities when provided with sufficient compute resources. Our trained smaller 1.5B parameter models, democratizes access to advanced AI capabilities for individuals, researchers, and organizations with limited computational resources. This accessibility is particularly valuable in educational settings where resource constraints often limit the adoption of large-scale AI systems. Our approach offers significant social benefits through its cost-effectiveness, reduced energy consumption, and lower computational requirements compared to larger models, making advanced reasoning capabilities available to a much wider audience. As shown in our analysis, tasks with low initial performance often exhibit sustained gains through extended training, creating opportunities to address reasoning challenges in critical domains like healthcare, climate science, and accessibility technologies. Small but powerful models can be deployed on-premises with enhanced security and privacy protections, making them suitable for sensitive applications in financial, legal, and healthcare sectors. Furthermore, these models' adaptability and lower latency make them ideal for real-time applications like AI teaching assistants, scientific research support, and specialized problem-solving tools that can significantly enhance human productivity across multiple domains.

**Ethical Considerations and Challenges**    Despite these opportunities, ProRL introduces important ethical considerations that require careful governance. The substantial training computational requirements may exacerbate resource inequality in AI development, while enhanced reasoning capabilities could enable more sophisticated misuse if deployed without appropriate safeguards. As these systems transition from no capability to high capability in certain reasoning tasks, ongoing monitoring

becomes essential to anticipate emergent behaviors and potential risks. Future work should combine ProRL techniques with explicit value alignment approaches, while developing dynamic evaluation benchmarks that evolve alongside model capabilities to ensure comprehensive assessment of both progress and risks across different contexts and communities.

# D   Example Prompts

## D.1   Graph Color Example

```
Graph Color Example:
Question: Please provide a coloring for this graph such that every vertex is
not connected to a vertex of the same color. The graph has these properties:

Vertices: [0, 1, 2, 3, 4, 5, 6, 7, 8, 9]
Edges: [(0, 1), (0, 7), (0, 9), (1, 4), (2, 4), (3, 5),
       (3, 6), (6, 8), (7, 9)]
Possible colors: [1, 2, 3]

Return your solution as a JSON map of vertices to colors.
(For example: {"0": 1, "1": 2, "2": 3}.)
```

## D.2   Family Relationships Example

```
Family Relationships Example:
Question: John is married to Isabella. They have a child called Edward.
Edward is married to Victoria.

What is Isabella to Edward? Respond only with the word that describes
their relationship.
```

## D.3   Boxnet Example

```
Question:
You are a central planner tasked with directing agents in a grid-like field
to move colored boxes to their corresponding color-coded targets.
Each agent occupies a 1x1 square and can only interact with objects within
its square. Agents can move a box to an adjacent square or
directly to a target square of the same color. A square may contain multiple
boxes and targets. The squares are identified by their center
coordinates (e.g., square[0.5, 0.5]). Actions are formatted as:
move(box_color, destination), where box_color is the color of the box and
destination is either a target of the same color or an adjacent square.
Your objective is to create a sequence of action plans that instructs
each agent to match all boxes to their color-coded targets in the most
efficient manner.

Please adhere to the following rules when specifying your action plan:
1. Single Action per Agent: Assign only one action to each agent at a time.
   However, the final answer shoule be a list of action plans for multiple steps.
2. Unique Agent Keys: Use unique keys for each agent in the JSON format action
   plan. The key should be the agent's coordinates in the format "Agent[x, y]".
3. Prioritize Matching Boxes to Targets: Always prioritize actions that will
   match a box to its target over moving a box to an adjacent square.
4. Sequential Action Planning: The whole returned answer should be a list of
   action plans for multiple steps, do not just return one step plan.
5. Clear Formatting: Ensure the action plan is clearly formatted in JSON, with
   each agent's action specified as a key-value pair.
6. Conflict Resolution: Ensure that no two agents are assigned actions that
   would interfere with each other.
```

```
7. Optimize Efficiency: Aim to minimize the number of moves required to match
   all boxes with their targets.

Here is the format for your action plan:
Please provide your final answer as a list of action dictionaries.
For example:
```json
[{"Agent[0.5, 0.5]": "move(box_blue, square[0.5, 1.5])",
"Agent[1.5, 0.5]": "move(box_red, target_red)"},
{"Agent[0.5, 1.5]": "move(box_blue, target_blue)",
"Agent[2.5, 0.5]": "move...}, {...}...]
```
Include an agent in the action plan only if it has a task to perform next.

The current left boxes and agents are:
Agent[0.5, 0.5]: I am in square[0.5, 0.5], I can observe
['box_red', 'target_red', 'box_blue', 'target_blue', 'box_green',
'target_green'], I can do ['move(box_red, square[0.5, 1.5])',
'move(box_red, target_red)', 'move(box_blue, square[0.5, 1.5])',
'move(box_blue, target_blue)', 'move(box_green, square[0.5, 1.5])',
'move(box_green, target_green)']
Agent[0.5, 1.5]: I am in square[0.5, 1.5], I can observe [], I can do []
```

# E   Training Dataset

We scale training across a wide spectrum of tasks that provide verifiable reward signals with details in Table 4. These tasks span from traditional reasoning domains, such as mathematical problem solving and code generation, to more complex and open-ended domains, including STEM-related problem solving, logical puzzles, and instruction following. The inclusion of such a diverse task set serves two key purposes. First, it broadens the model's exposure to a wide distribution of reasoning patterns, encouraging generalization beyond narrow, domain-specific behaviors. This is especially critical for developing models of adapting to new or unseen task formulations. Second, the task diversity enables a more rigorous evaluation of RL algorithms, as it tests their ability to learn robust decision-making strategies across fundamentally different environments and reward structures.

Table 4: Overview of training data used in our experiments, categorized by domain, reward type (binary or continuous), dataset size, and source. The datasets span a range of reasoning, coding, STEM, and instruction-following tasks.

| Data Type | Reward Type | Quantity | Data Source |
|---|---|---|---|
| Math | Binary | 40k | DeepScaleR Dataset |
| Code | Continuous | 24k | Eurus-2-RL Dataset |
| STEM | Binary | 25k | SCP-116K Dataset |
| Logical Puzzles | Continuous | 37k | Reasoning Gym |
| Instruction Following | Continuous | 10k | Llama-Nemotron |

## E.1   Math

We use high-quality, community-curated datasets made available through DeepScaleR [3]. The training set consists of 40K math problems sourced from a diverse range of national and international math competitions. We adopt DeepScaleR's original verifier and further augment it with an improved `math-verify`[3]. We obtain the LLM's answers by prompting the model with `Let's think step by step and output the final answer within \boxed{}`. We use a binary reward signal, assigning a score of 1 if the LLM's response passes either the original or the enhanced `math-verify`, and 0 otherwise (for incorrect or improperly formatted answers).

---

[3]`https://github.com/huggingface/Math-Verify`

### E.2 Code

We utilize publicly available reinforcement learning datasets comprising 24K coding problems [29], sourced from various programming competitions. To support continuous reward feedback, we improve code execution environment to run all test cases rather than terminating on the first error and assign rewards based on the fraction of test cases passed. Submissions that fail to compile, contain syntax errors, or exceed a 5 second total timeout are assigned a reward of zero. We also include instructions for the LLM to enclose its final code response with triple backticks.

### E.3 STEM

We use SCP-116K [60], a large-scale dataset containing 274k scientific problem-solution pairs spanning diverse fields such as physics, chemistry, biology, and mathematics. Each problem is accompanied by a corresponding solution extracted from the original source text, along with model-generated responses and reasoning paths produced by DeepSeek-R1. Given that SCP-116K was automatically extracted from heterogeneous and potentially noisy sources, we applied rigorous data filtering. First, we removed problems lacking a retrievable ground-truth solution from the source text. Then, we employed GPT-4o as a judge to assess whether the DeepSeek-R1 response aligned with the ground-truth answer. Only problems with consistent answers were retained, reducing the dataset from the original entries to 25K.

### E.4 Logical Puzzles (Reasoning Gym)

The logical puzzles are well-suited for reasoning model training due to their broad coverage of different reasoning skills, as well as their clear objectives and evaluation metrics. We utilize the Reasoning Gym project[4], which offers approximately 100 tasks across various domains, including algebra, arithmetic, computation, cognition, geometry, graph theory, logic, and popular games. To facilitate model training and evaluation, we generate a large dataset consisting of 37K synthetic training samples and 9600 validation samples, spanning 96 tasks. Notably, some tasks have a unique solution, whereas others, such as the Rubik's Cube and Countdown, admit multiple correct solutions. We employ the verifier provided by the Reasoning Gym repository for both model evaluation and reinforcement learning training signals. We use recommended default prompts which instruct models to enclose answers between `<answer>` `</answer>` tags.

### E.5 Instruction Following

To enhance our model's instruction-following capabilities, we leverage synthetic generated data from Llama-Nemotron [61] which data format is similar to IFEval [37]. Specifically the dataset contains synthetic prompts that pair tasks with randomly chosen instructions. For instance, a prompt may ask the model to "Write an essay about machine learning", while the instruction specifies, "Your response should have three paragraphs." We do not add further instructions on formatting and obtain the models response after thinking (`</think>` token).

## F   Training Recipe

**Training Monitoring.** We construct a validation data blend to closely monitor training progress across steps. This validation set includes subsets from our evaluation benchmark, specifically AIME2024, Codeforces, GPQA-diamond, IFEval, and the logic puzzle *graph_color* from Reasoning Gym. We evaluate model performance using similar sampling parameters as in evaluation settings (other than we use the same context window as in training).

**Reference Model and Optimizer Reset.** Occasionally, we perform a hard reset of the reference model and optimizer, as described in Section 2.3.1, particularly when validation metrics significantly degrade or when improvements plateau. Interestingly, the hard reset not only restores training stability but also provides an opportunity to adjust training hyperparameters and introduce enhancements such as additional training data and reward shaping. Figure 8 presents KL divergence across training runs. The final training recipe comprises several sequential stages, described in the following.

---

[4]`https://github.com/open-thought/reasoning-gym`

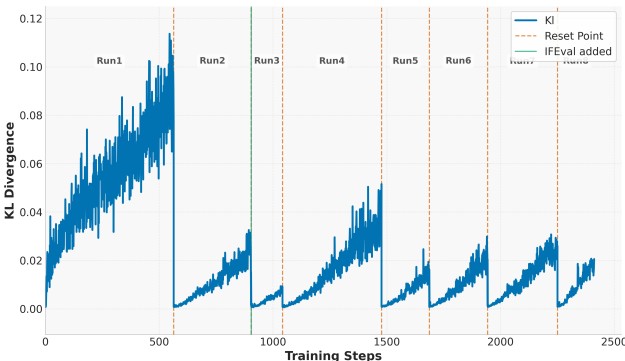

Figure 8: KL divergence across training runs. We periodically reset the reference policy and optimizer state during training.

- **Run 1:** We begin training on four tasks from Appendix E. We did not include instruction-following data as it was not available to us initially. In this phase, we limit the response length to 8k where the base model's sequential length is 128k to avoild long sequence rollouts. As shown in Figure 2, model response length first decreases shortly and then keeps increasing along with improved validation scores. Toward the end of this stage, we observe instability and degradation in validation performance.

- **Run 2:** We perform a hard reset of the reference policy and resume training with the same setup as Run 1. Unlike DeepScaleR [3], which proposes increasing the maximum response length, we maintain the maximum response length as 8k because we observe that 8k maximum length is sufficient for the model to learn and improve its validation scores.

- **Run 3:** We incorporate instruction-following data into the training mix and continue training. This stage proceeds until we observe a sudden increase in response length, primarily due to the model repeating answers and failing to terminate with an `<eos>` token.

- **Run 4 and 5:** We introduce reward shaping by penalizing responses that do not terminate correctly. This encourages proper generation behavior, resulting in a modest reduction in response length.

- **Runs 6 and 7:** We increase the rollout count from 16 to 32, performing two hard resets in the process. Interestingly, response length begins to rise again alongside improvements in validation metrics.

- **Run 8:** We extend the context window to 16k tokens and reduce rollout count to 16. Despite the model being trained on an 8k context window for most of the time, it quickly adapts to the extended context window. We observe marginal improvements in hard math tasks like AIME, with more substantial gains coming from other domains.

## G   Results Details

### G.1   Reasoning Gym

For logic puzzles in the Reasoning Gym suite, we adopt the categorization of 96 tasks as defined by the official GitHub repository. We show category performance details of our model in Table 5. Notably, DeepSeek-R1-Distill-Qwen-1.5B underperforms even on relatively simple mathematical tasks such as algebra and arithmetic. Closer inspection reveals that the model consistently formats its answers using \boxed{} rather than adhering to the instruction to use `<answer> </answer>` tags. Despite poor initial formatting behavior, the model is able to achieve high accuracy on these easier tasks post training, suggesting that formatting is relatively easy to learn. Our models still exhibit room for improvement on more challenging categories, including tasks from arc, code, cognition, and games. In these cases, the model often fails to make meaningful progress. Further analysis indicates that these failures stem from either a lack of core reasoning skills necessary to solve specific subtasks or insufficient background knowledge related to the problem domains. Addressing these limitations

Table 5: Detailed Reasoning Gym performance across all subcategories. Our model demonstrates superior performance across all reasoning tasks compared to both DeepSeek models.

| Reasoning Gym Performance (Part 1) | | | | | | | |
|---|---|---|---|---|---|---|---|
| Model | algebra | algorithmic | arc | arithmetic | code | cognition | games |
| DeepSeek-R1-Distill-Qwen-1.5B | 0.73 | 3.56 | 1.53 | 5.36 | 1.22 | 6.47 | 2.34 |
| DeepSeek-R1-Distill-Qwen-7B | 45.80 | 21.75 | **3.42** | 55.43 | 7.84 | 30.46 | 5.15 |
| Nemotron-Research-Reasoning-Qwen-1.5B | **97.21** | **53.90** | 2.52 | **82.81** | **29.84** | **40.16** | **26.38** |

| Reasoning Gym Performance (Part 2) | | | | |
|---|---|---|---|---|
| Model | geometry | graphs | induction | logic | Avg |
| DeepSeek-R1-Distill-Qwen-1.5B | 1.05 | 6.64 | 1.32 | 10.90 | 4.24 |
| DeepSeek-R1-Distill-Qwen-7B | 17.38 | 33.29 | 29.31 | 34.96 | 28.55 |
| Nemotron-Research-Reasoning-Qwen-1.5B | **89.84** | **66.49** | **73.50** | **82.94** | **59.06** |

may require additional finetuning data to better support model from a cold start, which we leave these enhancements to future work.

## G.2 Pass@k Comparisions

We share pass@k comparision plots across 3 models for all evaluated tasks. Due to compute resource limitations, we randomly select a subset of tasks from reasoning gym.

## G.3 Pass@1 Distribution Shifts

We share pass@1 distribution shifts for all evaluated tasks. Due to compute resource limitations, we randomly select a subset of tasks from reasoning gym.

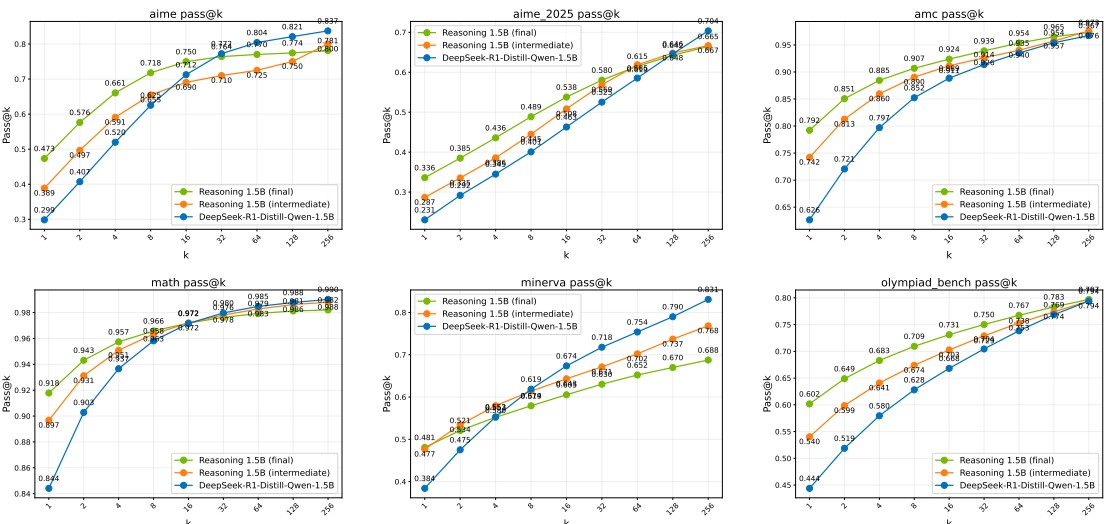

Figure 9: Pass@k for tasks in Math.

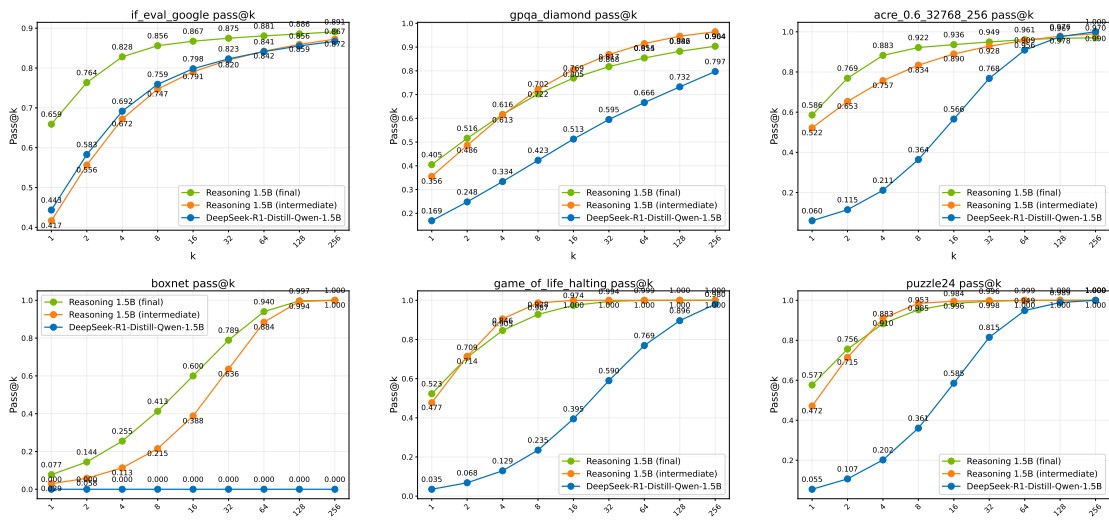

Figure 10: Pass@k for IFEval, GPQA, and Out-of-Distribution (OOD) tasks.

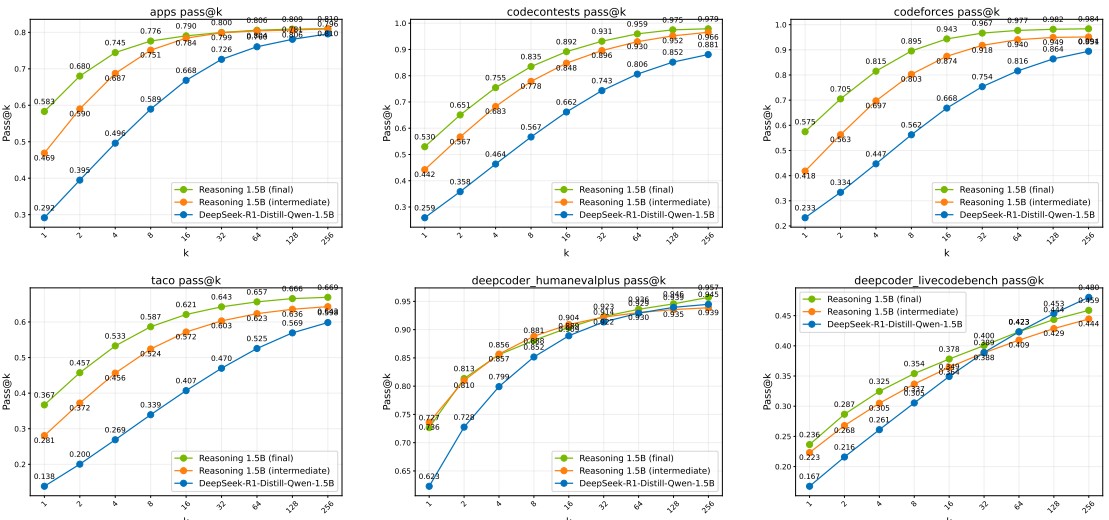

Figure 11: Pass@k for tasks in Code.

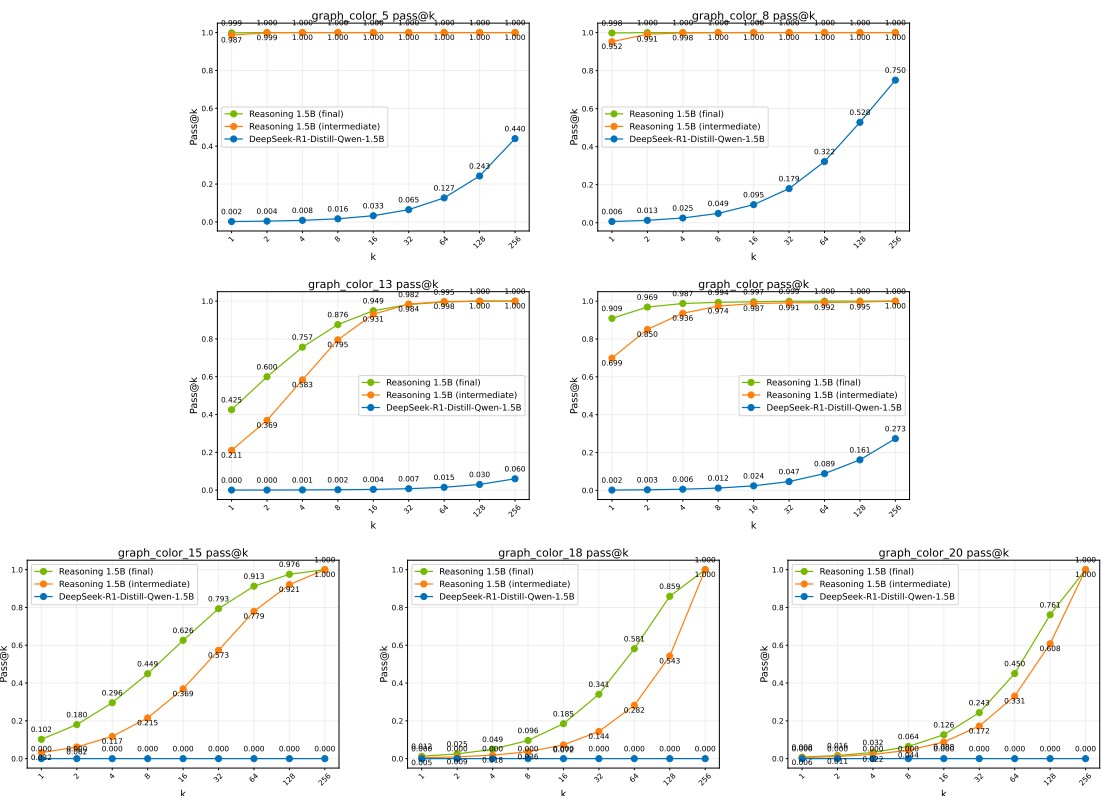

Figure 12: Pass@k for additional varying difficulty (number of nodes) in *graph_color*. Default used in training is 10 nodes.

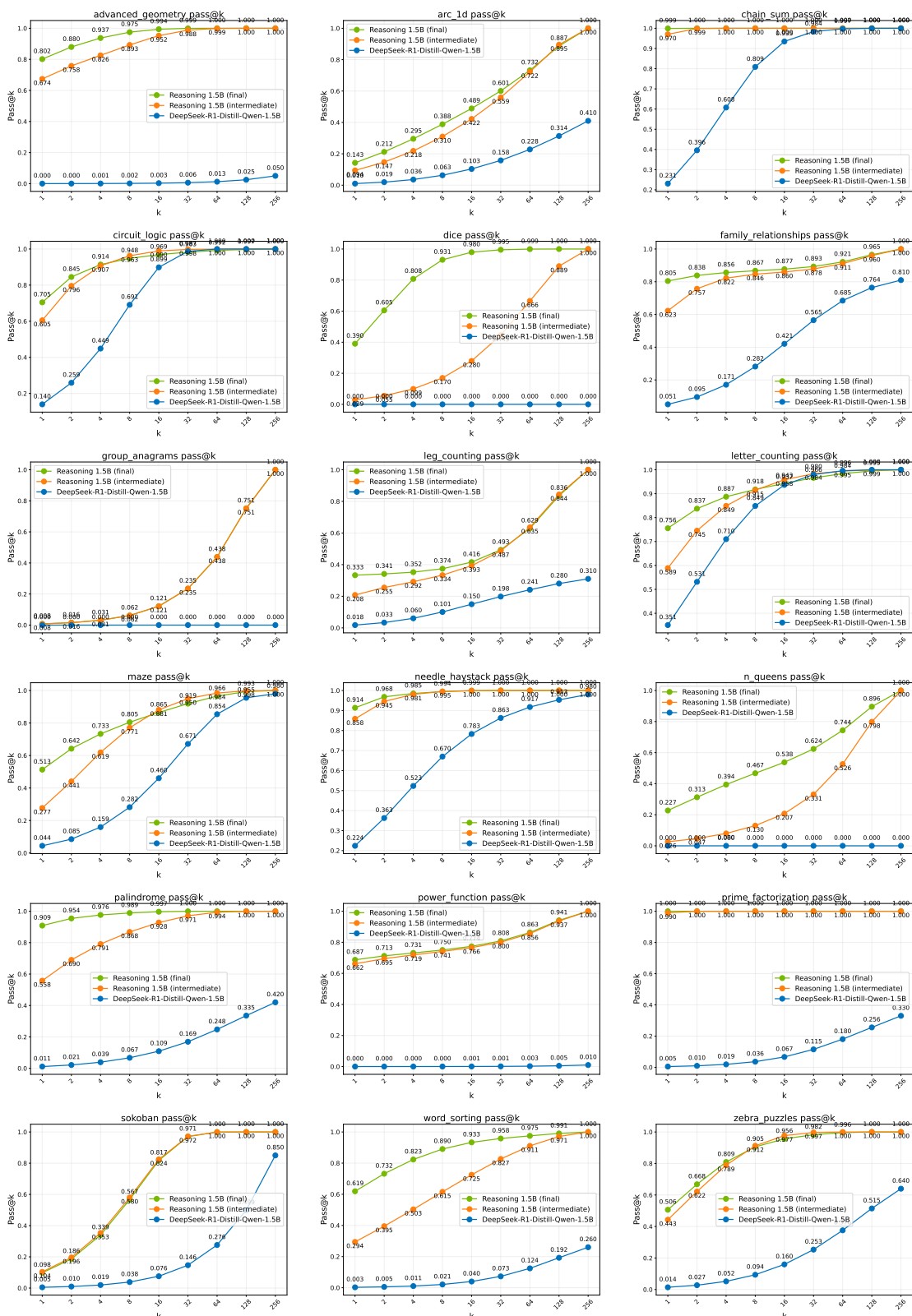

Figure 13: Pass@k for tasks in Reasoning Gym.

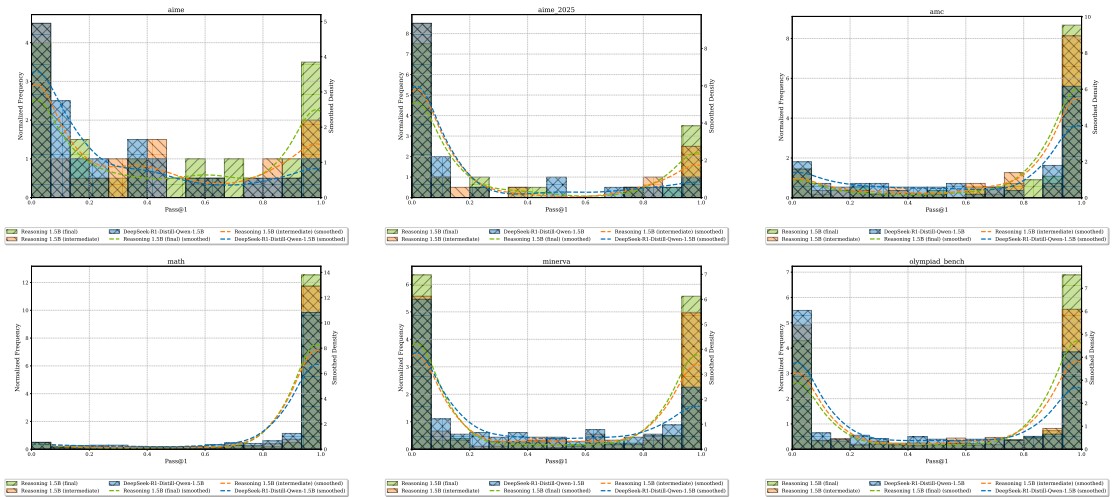

Figure 14: Pass@1 distribution for tasks in Math.

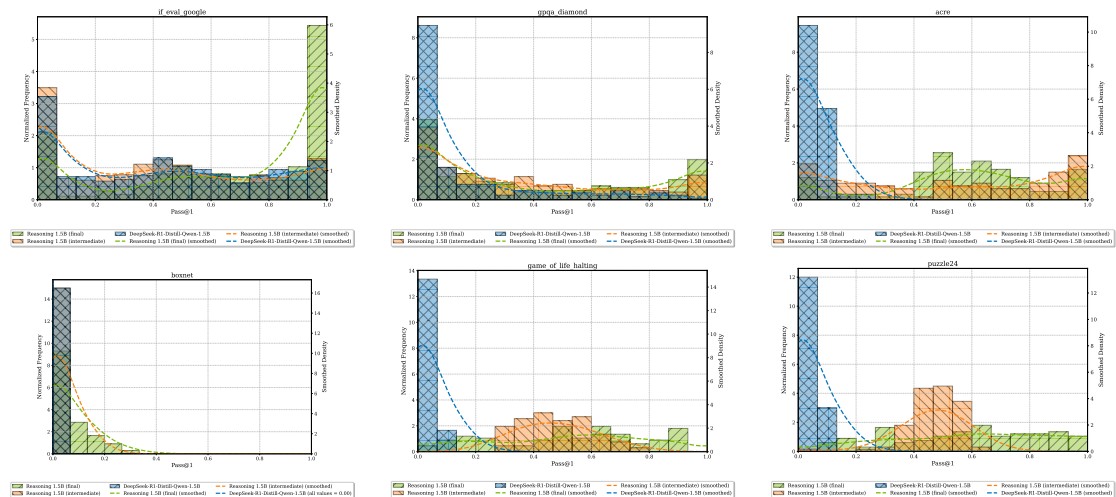

Figure 15: Pass@1 distribution for IFEval, GPQA, and Out-of-Distribution (OOD) tasks.

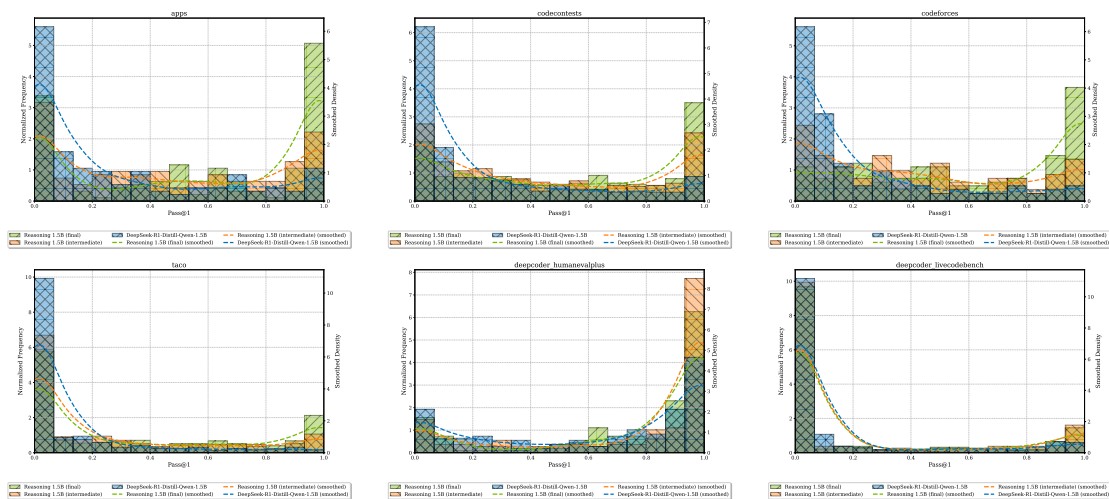

Figure 16: Pass@1 distribution for tasks in Code.

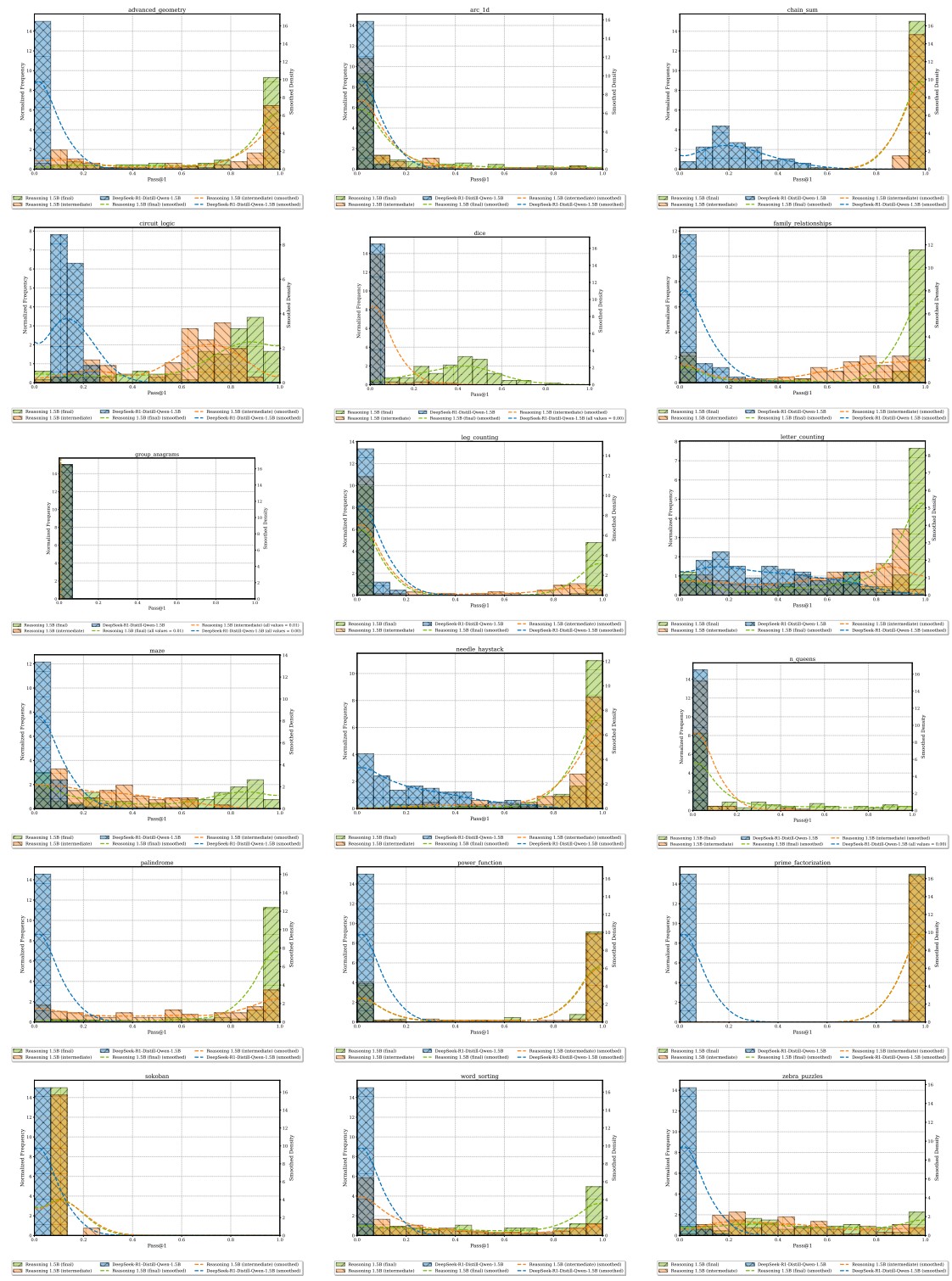

Figure 17: Pass@1 distribution for tasks in Reasoning Gym.

