# OpenReview forum: "ProRL: Prolonged Reinforcement Learning Expands Reasoning Boundaries in Large Language Models"
_NeurIPS.cc/2025/Conference — NeurIPS 2025 poster_

### Official Review · Reviewer_nT4B · 2025-06-28

**Clarity:** 3
**Significance:** 3
**Originality:** 2
**Rating:** 4
**Confidence:** 3

**Summary:**

This work introduces prolonged RL (ProRL), an optimization method that incorporates KL divergence control and reference policy resetting on policy optimization for LLMs. ProRL conducts its training over a diverse suite of tasks. Its empirical analysis shows that prolonged RL-trained models can consistently outperform the base model, suggesting that prolonged RL can discover new solution pathways that were not present in base models. This work also quantifies the model's novelty in reasoning via a creativity index for more quantitative analysis.

**Questions:**

Please see the concerns raised in the weakness section. I am happy to increase the score if the author is able to address my concerns.

**Ethical Concerns:**

["NO or VERY MINOR ethics concerns only"]

**Final Justification:**

The ablation study is important for this work and for readers to understand what components enable prolonged training with diverse explorations. Given the additional details and the promised improved manuscript with all additional ablation details, I am increasing my final rating to 4.

**Limitations:**

Yes.

**Paper Formatting Concerns:**

No.

**Quality:**

3

**Strengths And Weaknesses:**

Strength:
1. The findings delivered by this work are quite insightful and beneficial to the community for post-training LLMs. ProRL shows that with appropriate extropy collapse mitigation techniques, prolonged RL training on a diverse set of tasks can elicit new abilities from even SLMs.

2. The diverse and verifiable training dataset that covers 5 task domains also serves as a good contribution for eliciting SLMs reasoning capability outside of the previously dominating Math and Coding sets.

Weakness:
1. Though this work delivers useful findings on expanding the reasoning boundary, the source of the techniques that truly elicit new reasoning patterns is not analyzed thoroughly in this work. ProRL integrates KL regularization, Reference Policy Reset, and a diverse set of tasks as the training resources. However, the ablation for each component is not studied. Consequently, it is not clear from a reader's perspective whether the key ingredient came from the reference policy reset, maintaining a reasonable entropy during training, or the diversified task set appears to be more important.

2. The base model is limited to DeepSeek-R1-Distill-Qwen-1.5B. Recently, different base models have been discussed to exhibit completely different post-training behaviors. It is important to include another family of base models for conducting ProRL and with a larger model larger than 1.5B.

3. As described in weakness 1, the main results did not compare other training baselines, such as GRPO and DAPO, on the curated training datasets. It can also be beneficial to provide an ablation on the training data mixture.

---

> ### Author Rebuttal · Authors · 2025-07-30
>
> Thank you for the feedback and acknowledgement of our work. We hope the following responses could clarify your concerns.
>
>
> 1. *However, the ablation for each component is not studied.*
>
>
> We did conduct careful small scale ablation studies on our alternations of the GRPO pipeline, including decoupling  hyperparameters, adding reference model (and optimizer reset) hard reset. We apologize for not including such experiment details.
>
>
> **We will be providing an additional detailed technical report alongside the final version of the paper.** Unfortunately the conference review process prohibits sharing additional documents, thus we are only able to provide brief text descriptions on the ablation studies and key observations as follows:
>
>
> 1.1 Rollout Sampling Temperature
>
>
> Ablations with temperatures {0.6, 1.2} show that low temperature (0.6) causes early instability, reduced diversity, and overexploitation, hurting reward and validation. High temperature (1.2) yields steadier improvement, better generalization, and avoids mode collapse even in late-stage training, so we use 1.2 throughout.
>
>
> 1.2 Decoupled Clipping
>
>
> We vary ϵlow {0.1, 0.2} and ϵhigh​ {0.2, 0.3, 0.4}; ϵlow=0.2 better handles negative advantage, and ϵhigh=0.4 prevents entropy collapse, giving best validation. Smaller ϵlow increases entropy but not learning, so we adopt (0.2, 0.4).
>
>
> 1.3 Dynamic Sampling
>
>
> We further verified that dynamic sampling led to faster improvements than static sampling by filtering out prompts with zero advantage. This increases reward signal density in each batch, thereby improving the overall sample efficiency of training.
>
>
> 1.4 Resetting Reference Policy
>
>
> We observe that extending training with the same setting and training hyperparameters does not always lead to consistent improvements. We observed that by simply extending the training or Run1 (this is the DAPO baseline) without reference policy reset, it encounters a sharp drop in validation scores on the Codeforces benchmark. We were not able to scale RL training beyond 600 steps with DAPO, and in contrast resetting the reference policy allowed us to scale training to beyond 2k steps.
>
>
> 1.5 Mitigating Entropy Collapse
>
>
> We conducted a series of ablation studies evaluating the training dynamics under different strategies to mitigate entropy collapse. We investigated the following settings:
>
>
> - GRPO-KL=0: We employ standard GRPO training without KL divergence penalty or entropy bonus.
>
>
> - GRPO-KL=1e-4: We apply a small KL regularization term 1e-4 to control the divergence between the online policy and a reference policy. This helps prevent the model from drifting too far and encourages continued exploration.
>
>
> - DAPO: This is the DAPO baseline with decoupled clipping and dynamic sampling. No KL divergence penalty or entropy bonus.
>
>
> We observe that GRPO-KL=0 suffers from entropy collapse and the validation score stops improving after the entropy degrades to ~0 in ~500 steps. In contrast, DAPO and KL penalty both can mitigate entropy collapse as the validation scores continue to improve. Based on these findings, we adopt a combination of DAPO and a KL penalty in our final training setup.
>
>
> 2. *It is important to include another family of base models for conducting ProRL and with a larger model larger than 1.5B.*
>
>
> We conduct further experiments by applying our training recipe to a larger model and different base model. We use Llama-3.1-Nemotron-Nano-8B-v1, which is a derivative based on llama-3.1-8B-Instruct, further finetuned on diverse reasoning tasks. Due to time constraints and compute limitations, we only scale training to ~1.4k steps. We provide detailed comparison with the base model across tasks as follows
> | Model                         | Math  | Code  | GPQA  | IFEval | Reasoning Gym |
> |-------------------------------|-------|-------|-------|--------|---------------|
> | Llama-3.1-Nemotron-Nano-8B-v1 | 68.26 | 61.20 | 52.76 |  66.88 |     12.15     |
> | ProRL-8B                      | 71.98 | 72.80 | 55.52 |  70.94 |     77.23     |
>
>
> Note: The above reported results on Code only consists of {apps, condecontests, codeforces, taco} and report the obtained continuous reward. All other benchmarks are the same as in the paper.
>
>
> 3. *The main results did not compare other training baselines, such as GRPO and DAPO, on the curated training datasets.*
>
>
> We apologize again for not including detailed ablation studies in the original manuscript. We hope our response to the 1st question could provide confidence in our results when comparing against other training baselines such as GRPO and DAPO. In our ablation studies, we found GRPO performance to saturate within ~600 steps, largely due to entropy collapse as the model loses the capability to explore. We also found DAPO without reference policy reset to have training stability issues, in having some validation scores degrading ~500 steps.
>
>
> As requested from Reviewer TRB6, we further conducted ablation studies comparing with other finetuning techniques such as SFT and DPO. We decided to focus on tasks within the Reasoning Gym suite and used a stronger base model DeepSeek-R1-Distilled-Qwen-32B model to generate reasoning traces and used verifiers to filter negative samples. We constructed a finetuning dataset of ~200k verified responses. We further collected additional negative responses across the same dataset, constructing a preference dataset of ~200k entries for DPO finetuning. We conduct SFT and DPO for 2 epochs using DeepSeek-R1-Distilled-Qwen-1.5B as the base model. We provide comparisons with these additional baselines with our ProRL-1.5B model on the Reasoning Gym tasks as follows:
> |       Model      | algebra | algorithmic |  arc | arithmetic |  code | cognition | games | geometry | graphs | induction | logic |  Avg  |
> |:----------------:|:-------:|:-----------:|:----:|:----------:|:-----:|:---------:|:-----:|:--------:|:------:|:---------:|:-----:|:-----:|
> | DeepSeek-R1-1.5B |   0.73  |     3.56    | 1.53 |    5.36    |  1.22 |    6.47   |  2.34 |   1.05   |  6.64  |    1.32   | 10.90 |  4.24 |
> |  ProRL-1.5B (2k) |  97.21  |    53.90    | 2.52 |    82.81   | 29.84 |   40.16   | 26.38 |   89.84  |  66.49 |   73.50   | 82.94 | 59.06 |
> |     SFT-1.5B     |  51.18  |    25.65    | 1.48 |    64.18   |  1.35 |   39.13   | 17.64 |   65.13  |  35.63 |   60.59   | 41.21 | 35.81 |
> |     DPO-1.5B     |  48.61  |    24.29    | 1.53 |    62.76   |  0.88 |   37.37   | 15.64 |   64.12  |  32.92 |   58.47   | 38.63 | 34.15 |
>
>
> 4. *It can also be beneficial to provide an ablation on the training data mixture.*
>
>
> In regards to training data mixture, we conduct a small scale experiment with the following two data mixtures: Math-Only, consisting of only the math dataset; and Math-Code-STEM-Reasoning, consisting of the math, code, stem reasoning and logic puzzle (Reasoning Gym) data. We only train for ~600 steps with the comparison as follows:
> | Model                         | Math  | Code  | GPQA  | Reasoning Gym |
> |-------------------------------|-------|-------|-------|---------------|
> | Base Model | 44.45 | 14.98 | 15.86 | 4.24     |
> | Math-Only | 53.37 | 17.21 | 17.48 |  4.51     |
> | Math-Code-Stem-Reasoning | 53.45 | 24.57 | 37.12 |  49.87   |
>
>
> Note: The above reported results on Code only consists of pass@1 for {apps, condecontests, codeforces, taco}.
>
>
> As shown in the above, training only on math data does bring some level of transfer to other tasks but not as significant as directly including similar tasks in training. This highlights the importance of training on a diverse mixture of data.

---

> > ### Comment · Reviewer_nT4B · 2025-08-05
> >
> > I thank the authors for the insightful rebuttal. Given the additional details and the promised improved manuscript with all additional ablation details, I have increased my rating.

---

### Official Review · Reviewer_SYom · 2025-06-30

**Clarity:** 3
**Significance:** 4
**Originality:** 4
**Rating:** 5
**Confidence:** 4

**Summary:**

This paper sheds some light on the training dynamics of GRPO for reasoning models. The authors present a really comprehensive study where they first identify problems in the training dynamics due to entropy collapse during prolonged training. Previous literature have used KL regularizers to mitigate this issue, however, the authors find that this only slows the occurrence of this issue. Therefore, they propose an hard reset of the reference policy which is useful to smooth down the training. The authors apply this new training recipe to the base model DeepSeek-R1-1.5B and are able to substantially outperform it on several benchmarks.

This paper represents a great contribution to the community and provides important insights into current best practices in using RL for LLMs.

**Questions:**

1. I was wondering how the solution proposed by this paper compares with current real-world implementations of GRPO such as TRL (from Huggingface). For instance, they explicitly mention the following:


> Note that compared to the original formulation in DeepSeekMath: Pushing the Limits of Mathematical Reasoning in Open Language Models, we use β = 0.0 by default, meaning that the KL divergence term is not used. This choice is motivated by several recent studies (e.g., Open-Reasoner-Zero: An Open Source Approach to Scaling Up Reinforcement Learning on the Base Model) which have shown that the KL divergence term is not essential for training with GRPO. As a result, it has become common practice to exclude it (e.g. Understanding R1-Zero-Like Training: A Critical Perspective, DAPO: An Open-Source LLM Reinforcement Learning System at Scale).

2. How do you determine when the model stagnates? Do you have a way to determine this automatically? This would be interesting to understand more deeply, considering that it might have an effect on how these results can generalise to other model families.

3. I find Figure 4 a bit confusing. Specifically, I'm not sure how to concretely read it considering that all the trends are very similar across the different categories.

4. Did you notice any specific difference in performance between tasks that have binary or continuous rewards?

**Ethical Concerns:**

["NO or VERY MINOR ethics concerns only"]

**Final Justification:**

I think this is a great paper which explores a really important research question in the space of reasoning models. I thank the reviewers for providing additional details on this paper. I think that the authors should focus their attention on reporting these details in their manuscript to clearly highlight the main contribution and the scope of the work. For instance, I think that it's important to highlight that this work focuses on further improving a reasoning model, so it assumes that the starting point of their training regime is already quite good. Additionally, reporting more details regarding the training and how they monitor the training is essential. I would suggest the authors implement this in two ways: 1) release your code with clear indications regarding how to reproduce your training runs; 2) release some logbooks that showcase when the authors detected specific faulty runs or situations where their model was stagnating.

**Limitations:**

yes

**Quality:**

4

**Strengths And Weaknesses:**

### Strengths

1. Really well-written paper which includes a detailed background as well as a detailed analysis of the experiments.
2. The authors present an interesting training regime that mitigates important issues in current RL training for reasoning models with a particular focus on entropy collapse.
3. The training regime is applied only to DeepSeek-R1 and showcases an impressive boost in performance not only on mathematical reasoning but also on code, instruction following and logic puzzles. In my opinion, this represents the most important contribution of this paper.
4. The paper provides a lot of insights into the training dynamics of these training regimes. Particularly, they study how ProRL improves the performance and the ability to elicit new reasoning patterns.

### Weaknesses

I really liked the paper and I think it represents an important contribution for the community. However, I would like to point out some weaknesses that this work might have and that I believe might help researchers make more informed decisions in the future. I list them below:

1. It's not clear what the impact of the selection of the base model is. At the moment, the authors start from DeepSeek-R1, which presumably is already skilled in producing reasoning chains. I appreciate that running these experiments is a big endeavour and I wonder whether the authors have explored what happens when starting from a base instruction-tuned model like Llama or Qwen.

2. It is not clear to me how the authors determine when the model stagnates. Based on Section 3.3, this seems to be a manual process and therefore might be very model-specific. I think it would be useful to report this point in the limitations or, in general, it would be useful to be explicit about the limitations of this training regime.

---

> ### Author Rebuttal · Authors · 2025-07-30
>
> Thank you for the feedback and acknowledgement of our work. We hope the following responses could clarify your concerns.
>
>
> 1. *I wonder whether the authors have explored what happens when starting from a base instruction-tuned model like Llama or Qwen.*
>
>
> Thank you for the question. We primarily wanted to experiment whether RL can bring improvements to already strong reasoning models. We thus leave research on scaling RL from non-reasoning models as an open question for future research.
>
>
> We conduct further experiments by applying our training recipe to a larger model and different base model. We use Llama-3.1-Nemotron-Nano-8B-v1, which is a derivative based on llama-3.1-8B-Instruct, further finetuned on diverse reasoning tasks. Due to time constraints and compute limitations, we only scale training to ~1.4k steps. We provide detailed comparison with the base model across tasks as follows
> | Model                         | Math  | Code  | GPQA  | IFEval | Reasoning Gym |
> |-------------------------------|-------|-------|-------|--------|---------------|
> | Llama-3.1-Nemotron-Nano-8B-v1 | 68.26 | 61.20 | 52.76 |  66.88 |     12.15     |
> | ProRL-8B                      | 71.98 | 72.80 | 55.52 |  70.94 |     77.23     |
>
>
> Note: The above reported results on Code only consists of {apps, condecontests, codeforces, taco} and report the obtained continuous reward. All other benchmarks are the same as in the paper.
>
>
> 2. *It is not clear to me how the authors determine when the model stagnates. Based on Section 3.3, this seems to be a manual process and therefore might be very model-specific. I think it would be useful to report this point in the limitations or, in general, it would be useful to be explicit about the limitations of this training regime.*
>
>
> We determine stagnation through manual inspection guided by quantitative signals. Specifically, we reset the reference model when (i) the KL loss exceeds a threshold (e.g., 0.2), indicating excessive policy drift, or (ii) validation performance shows a consistent downward trend. In such cases, we resume from the most recent checkpoint with the best validation score. We will explicitly note in the limitations section that this process involves manual intervention and may be model-specific.
>
>
> 3. *I was wondering how the solution proposed by this paper compares with current real-world implementations of GRPO such as TRL (from Huggingface).*
>
>
> I believe both the work mentioned (Open-Reasoner-Zero and DAPO) mainly experiment with using pretraining checkpoints (no instructional finetuning) as the base model. In our case, we select already strong reasoning models as the base model. In the case of starting from a weak model, removing KL penalty could facilitate the model learning as this additional penalty term constrains the policy term potentially limiting exploration. In our case, our ablation studies still found small KL divergence to be beneficial when starting from a strong base model, since it mitigates the entropy collapse issue and enables scaling training to thousands of steps. The reference policy resets ensures stability, and also allows the policy model also to gradually diverge from the base model.
>
>
> Most popular RL frameworks (such as TRL and verl) still keep KL divergence penalty coefficient as a tunable parameter, allowing easy application of our proposed method.
>
>
> 4. *Did you notice any specific difference in performance between tasks that have binary or continuous rewards?*
>
>
> Thank you for the great question! We do notice differences on tasks such as coding with binary and continuous rewards. Using continuous reward during initial training stages improves the model learning since these partial rewards help guide weaker models to achieve some level of competence. Furthermore, since dynamic sampling filters instances with pass rate of 0/1, continuous reward would improve efficiency as instances with intermediate rewards still have non-zero advantage. However, we also observe that during the late stages, switching to binary rewards increases the pass@1 score (but lowers the continuous reward scores) of the model performance. Nonetheless we keep tasks to continuous reward as to keep consistency during training.

---

> > ### Comment · Reviewer_SYom · 2025-08-04
> >
> > Thank you for clarifying the points above and providing some preliminary results on Nemontron. Please see my final justification for additional details and comments.

---

### Official Review · Reviewer_Dkqj · 2025-07-03

**Clarity:** 3
**Significance:** 4
**Originality:** 3
**Rating:** 5
**Confidence:** 3

**Summary:**

This work examines reinforcement learning for reasoning and attempts to show that it does indeed genuinely extend the base model's reasoning ability. The authors goal is to question a number of recent papers which claim that reinforcement learning merely improves search of the existing base model's reasoning space.  The authors show that there are a number of methodological problems with these other papers (they have overfit on pretraining/finetuning and their RL and crucially they have terminated prematurely) They then propose a prolonged RL recipe ProRL which fixes these problems.  Elements of this recipe include diverse training data, and mitigating entropy collapse with high sampling temperature and more flexible clipping of the PPO objective, and finally KL regularisation with hard resets.  They then show that with this recipe they can train in a stable fashion for much longer, reaching up to 2000 steps. The have extensive experiments where they show that with ProRL they can increase the boundary of the model's reasoning ability (as measured by accuracy @ 128). They show improvements in overall reasoning accuracy across many tasks covering maths, code and stem problems. The provide interesting analyses on when this extended RL training helps and when it does not.

**Questions:**

Questions
	1) DeepScaler, DeepCoder in Tables 1 and 2 are not explained.
	2) I suggest you use the same colours for the models in both Figure 5 and Figure 6 for easing understanding - a somewhat ironic suggestion for a task called "graph colour" :)

**Ethical Concerns:**

["NO or VERY MINOR ethics concerns only"]

**Limitations:**

Yes - Limitations were well described and indeed there are a number of significant limitations including limited model size - we don't know how well this will scale to larger models.

**Quality:**

3

**Strengths And Weaknesses:**

Strengths:
	1) The authors identify problems with existing research methodology which claims that RL does not increase the model's ability to reason.
	2) They propose a recipe that allows RL to provide improvements throughout prolonged RL training.
	3) They have extensive experiments showing the benefit of their recipe.
	4) They do analysis that show what types of conditions are necessary for prologued training to help (difficult tasks) or hinder (simpler or overfitted tasks)

Weaknesses:
	1) No ablation studies - what part of the recipe made all the difference in performance? Are all pieces equally necessary? At least some discussion of this is warrented.
	2) The methods proposed are not particularly novel - however the value of the paper is in the insight into the problems in methodology of other RL papers, and finding a recipe that fixes these problems. I do not this this is a major flaw.
	3) Why stop at 2000? Figure 1 shows continued improvement although I am sure you had a stopping criteria. Is further exploration of the reasoning space impossible? Some discussion of the limits would be good.
	4) How do you know if your task is going to benefit from extended RL? Would you need task specific stopping criteria? This does not seem ideal if you have many different tasks with different inherent complexities.

---

> ### Author Rebuttal · Authors · 2025-07-30
>
> Thank you for the feedback and acknowledgement of our work. We hope the following responses could clarify your concerns.
>
>
> 1. *No ablation studies.*
>
>
> We did conduct careful small scale ablation studies on our alternations of the GRPO pipeline, including decoupling  hyperparameters, adding reference model (and optimizer reset) hard reset. We apologize for not including such experiment details.
>
>
> **We will be providing an additional detailed technical report alongside the final version of the paper.** Unfortunately the conference review process prohibits sharing additional documents, thus we are only able to provide brief text descriptions on the ablation studies and key observations as follows:
>
>
> 1.1 Rollout Sampling Temperature
>
>
> Ablations with temperatures {0.6, 1.2} show that low temperature (0.6) causes early instability, reduced diversity, and overexploitation, hurting reward and validation. High temperature (1.2) yields steadier improvement, better generalization, and avoids mode collapse even in late-stage training, so we use 1.2 throughout.
>
>
> 1.2 Decoupled Clipping
>
>
> We vary ϵlow {0.1, 0.2} and ϵhigh​ {0.2, 0.3, 0.4}; ϵlow=0.2 better handles negative advantage, and ϵhigh=0.4 prevents entropy collapse, giving best validation. Smaller ϵlow increases entropy but not learning, so we adopt (0.2, 0.4).
>
>
> 1.3 Dynamic Sampling
>
>
> We further verified that dynamic sampling led to faster improvements than static sampling by filtering out prompts with zero advantage. This increases reward signal density in each batch, thereby improving the overall sample efficiency of training.
>
>
> 1.4 Resetting Reference Policy
>
>
> We observe that extending training with the same setting and training hyperparameters does not always lead to consistent improvements. We observed that by simply extending the training or Run1 (this is the DAPO baseline) without reference policy reset, it encounters a sharp drop in validation scores on the Codeforces benchmark. We were not able to scale RL training beyond 600 steps with DAPO, and in contrast resetting the reference policy allowed us to scale training to beyond 2k steps.
>
>
> 1.5 Mitigating Entropy Collapse
>
>
> We conducted a series of ablation studies evaluating the training dynamics under different strategies to mitigate entropy collapse. We investigated the following settings:
>
>
> - GRPO-KL=0: We employ standard GRPO training without KL divergence penalty or entropy bonus.
>
>
> - GRPO-KL=1e-4: We apply a small KL regularization term 1e-4 to control the divergence between the online policy and a reference policy. This helps prevent the model from drifting too far and encourages continued exploration.
>
>
> - DAPO: This is the DAPO baseline with decoupled clipping and dynamic sampling. No KL divergence penalty or entropy bonus.
>
>
> We observe that GRPO-KL=0 suffers from entropy collapse and the validation score stops improving after the entropy degrades to ~0 in ~500 steps. In contrast, DAPO and KL penalty both can mitigate entropy collapse as the validation scores continue to improve. Based on these findings, we adopt a combination of DAPO and a KL penalty in our final training setup.
>
>
> 2. *Why stop at 2000 steps.*
>
>
> We stopped at 2k steps largely due to resource limitations and time constraints. We were able to further scale training for an additional 1k steps to a total 3k steps. We compare the new model with the original 2k model in our paper as follows:
> | Model                         | Math  | Code  | GPQA  | IFEval | Reasoning Gym |
> |-------------------------------|-------|-------|-------|--------|---------------|
> | ProRL-2k  | 60.14 | 37.49 | 41.78 |  66.02 |     59.06     |
> | ProRL-3k  | 61.69 | 41.03 | 41.32 |  70.85 |     62.49     |
>
>
> As shown above, prolonged RL continues to bring improved performance across most tasks. Since we largely observe log-linear scaling, linear growth in the performance translates to exponential training computation. Furthermore data might be another limitation, as dynamic sampling will filter easy tasks (with pass rate of 1). We also provided a Limitations section in the Appendix.
>
>
> 3. *How do you know if your task is going to benefit from extended RL? Would you need task specific stopping criteria?*
>
>
> In our experimental observations, all five tasks are generally suitable for ProRL. However, some tasks—such as those related to Code—show more noticeable and consistent improvement, while tasks in the MATH category improve more slowly. We do not impose a stopping criteria, as dynamic sampling removes too easy and too difficult problems from training. We do agree that having a stopping criteria (such as if the validation scores reach beyond a certain threshold) would greatly improve the efficiency of dynamic sampling.
>
>
> 4. *DeepScaler, DeepCoder in Tables 1 and 2 are not explained.*
>
>
> We included text descriptions under "Comparison with Domain-Specialized Mdoels”:
> “We compare the performance of NemotronResearch-Reasoning-Qwen-1.5B with two domain-specialized baselines: DeepScaleR-1.5B [3], tailored for mathematical reasoning, and DeepCoder-1.5B [7], focused on competitive programming tasks.”
> We will also include the related citations in the Table for clarity.
>
>
> 5. *I suggest you use the same colours for the models in both Figure 5 and Figure 6 for easing understanding*
>
>
> Thank you for pointing this out. We will keep colours of models consistent for Figure 5 and 6 in the final version.

---

> > ### Comment · Reviewer_Dkqj · 2025-08-05
> > **response to rebuttal**
> >
> > Thank you for your extensive rebuttal. I am happy that you have indeed got evidence for the value of the individual components of the ProRL recipe. This will hopefully strengthen the paper. You have addressed my other points.

---

### Official Review · Reviewer_TrB6 · 2025-07-17

**Clarity:** 3
**Significance:** 2
**Originality:** 3
**Rating:** 4
**Confidence:** 4

**Summary:**

Authors demonstrate that with prolonged tuning LM using RL with verifiable rewards can actually lead to better generalization in a wide range of domains. Tuned model demonstrates higher performance, bigger amount of novel solutions and better generalizability to OOD tasks.

**Questions:**

1) Please address as many weaknesses as possible. I hope I provided enough guidance for that. The core one is the weakness (1).
2) Figure 1 (middle) demonstrates that base model produces some amount of outliers which score much higher in terms of creativity compared to samples after RL finetuning. What are those outliers? Are those just random garbage? And do you have any hypothesis why we don't see those for RL model?
3) Is any weight decay used? AdamW is used as optimizer, however no weight decay parameter is mentioned. I would also recommend to report all the hyperparamters in the form of a table in Appendix and not as parts of the text.

**Ethical Concerns:**

["NO or VERY MINOR ethics concerns only"]

**Final Justification:**

Authors resolved most of my concerns

**Limitations:**

Yes

**Paper Formatting Concerns:**

-

**Quality:**

2

**Strengths And Weaknesses:**

# Strengths
1) Authors provide recipe for stable prolonged RL finetuning which includes KL regularization with hard update of the reference model, decoupled $\epsilon$ parameters for GRPO objective and specific schedule which involves various alterations over the training process.
2) Tuned model significantly outperforms base model on a wide range of tasks and better adapts to the OOD problems. It is worth mentioning that tuned 1.5B model (based on 1.5B DeepSeek-R1) works on-par with 7B DeepSeek-R1.
3) The work challenges claims from recent works which say that RL does not actually improve reasoning capabilities of the base model but only improves sampling. This work shows that in domains different from math this might not be the case.

# Weaknesses
1) The main goal of the work is to demonstrate that tuning LMs using RL with verifiable rewards is a good methodology. And it is demonstrated that for the challenging setups (where the base model struggled) RL tuning actually helps. I personally would be much more surprised if it was not the case. However, authors do not provide any additional baselines. If I understand correctly, base model was not trained on the data which is used for finetuning. But what if improvements come from just seeing the novel tasks during finetuning? I think that adding experiments with other tuning techniques is a **must-have** part of such research. E.g. using supervised tuning if reference reasoning trails are available for benchmarks, running language modeling over just the input tasks if reference solutions are not available or generating reasoning samples from stronger model, using DPO with things mentioned before. I doubt that supervised tuning or DPO with limited samples will demonstrate that strong performance, however those are more stable and much less cheaper in terms of compute. Anyway, I will struggle a lot to recommend this paper for publication without such baselines.
2) No ablation study is presented for the proposed approach. Authors alternate several things in GRPO pipeline such as decoupling $\epsilon$ hyperparameters, adding reference model (and optimizer reset) hard reset and non-trivial changes over the course of the training. That many changes should be carefully analyzed so further works could use pipeline confidently. No analysis on hyperparameters sensitivity as well.
3) No analysis on the various model sizes. While I understand that it potentially requires a lot of compute it would be nice to see the same behavior over varying base model sizes, e.g. DeepSeek-R1 7B and 14B.
4) It would be also interesting to see whether RL saturates at some point or not. Figure 1 (left) demonstrates that performance improvement requires exponential training time scale. But I would highly appreciate to see further tuning results (starting from the latest available checkpoint) if authors have resources for that.
5) (Formating) Font sizes should be increased for the most of the figures. It is hard to read figures now.

---

> ### Author Rebuttal · Authors · 2025-07-30
>
> Thank you for your review and suggestions. We hope the following responses could clarify your concerns.
>
>
> 1. *Additional baseline request such as SFT or DPO*
>
>
> Thank you for the suggestion. Our main contribution is to provide empirical analysis in demonstrating the benefits of prolonged reinforcement learning in expanding the reasoning boundary and uncovering novel reasoning strategies inaccessible to the base model. Such results are in fact not as straightforward, especially works targeting RLVR, since numerous concurrent works have argued otherwise. We hope our paper can bring forth more discussion within the research community on the benefits (or lack thereof) of prolonged RL on LLMs.
>
>
> We do agree that providing additional baselines would greatly strengthen our argument on the increased benefit of scaling Reinforcement Learning. Since the concern seems to be particularly centered on novel tasks outside the capabilities of the base model, we decide to focus on tasks within the Reasoning Gym suite. As per your recommendations, we used a stronger base model DeepSeek-R1-Distilled-Qwen-32B model to generate reasoning traces and used verifiers to filter negative samples. We constructed a finetuning dataset of ~200k verified responses. We further collected additional negative responses across the same dataset, constructing a preference dataset of ~200k entries for DPO finetuning. We conduct SFT and DPO for ~2 epochs using DeepSeek-R1-Distilled-Qwen-1.5B as the base model. We will make sure to include these additional baselines in our paper and details on the training setups etc. We provide only the following results for brevity:
>
>
> |       Model      | algebra | algorithmic |  arc | arithmetic |  code | cognition | games | geometry | graphs | induction | logic |  Avg  |
> |:----------------:|:-------:|:-----------:|:----:|:----------:|:-----:|:---------:|:-----:|:--------:|:------:|:---------:|:-----:|:-----:|
> | DeepSeek-R1-1.5B |   0.73  |     3.56    | 1.53 |    5.36    |  1.22 |    6.47   |  2.34 |   1.05   |  6.64  |    1.32   | 10.90 |  4.24 |
> |  ProRL-1.5B (2k) |  97.21  |    53.90    | 2.52 |    82.81   | 29.84 |   40.16   | 26.38 |   89.84  |  66.49 |   73.50   | 82.94 | 59.06 |
> |     SFT-1.5B     |  51.18  |    25.65    | 1.48 |    64.18   |  1.35 |   39.13   | 17.64 |   65.13  |  35.63 |   60.59   | 41.21 | 35.81 |
> |     DPO-1.5B     |  48.61  |    24.29    | 1.53 |    62.76   |  0.88 |   37.37   | 15.64 |   64.12  |  32.92 |   58.47   | 38.63 | 34.15 |
>
>
>
>
>
>
>
>
> 2. *No ablation study is presented for the proposed approach.*
>
>
> We did conduct careful small scale ablation studies on our alternations of the GRPO pipeline, including decoupling  hyperparameters, adding reference model (and optimizer reset) hard reset. We apologize for not including such experiment details.
>
>
> **We will be providing an additional detailed technical report alongside the final version of the paper.** Unfortunately the conference review process prohibits sharing additional documents or figures, thus we are only able to provide brief text descriptions on the ablation studies and key observations as follows:
>
>
> 2.1 Rollout Sampling Temperature
>
>
> Ablations with temperatures {0.6, 1.2} show that low temperature (0.6) causes early instability, reduced diversity, and overexploitation, hurting reward and validation. High temperature (1.2) yields steadier improvement, better generalization, and avoids mode collapse even in late-stage training, so we use 1.2 throughout.
>
>
> 2.2 Decoupled Clipping
>
>
> We vary ϵlow {0.1, 0.2} and ϵhigh​ {0.2, 0.3, 0.4}; ϵlow=0.2 better handles negative advantage, and ϵhigh=0.4 prevents entropy collapse, giving best validation. Smaller ϵlow increases entropy but not learning, so we adopt (0.2, 0.4).
>
>
> 2.3 Dynamic Sampling
>
>
> We further verified that dynamic sampling led to faster improvements than static sampling by filtering out prompts with zero advantage. This increases reward signal density in each batch, thereby improving the overall sample efficiency of training.
>
>
> 2.4 Resetting Reference Policy
>
>
> We observe that extending training with the same setting and training hyperparameters does not always lead to consistent improvements. We observed that by simply extending the training or Run1 (this is the DAPO baseline) without reference policy reset, it encounters a sharp drop in validation scores on the Codeforces benchmark. We were not able to scale RL training beyond 600 steps with DAPO, and in contrast resetting the reference policy allowed us to scale training to beyond 2k steps.
>
>
> 2.5 Mitigating Entropy Collapse
>
>
> We conducted a series of ablation studies evaluating the training dynamics under different strategies to mitigate entropy collapse. We investigated the following settings:
>
>
> - GRPO-KL=0: We employ standard GRPO training without KL divergence penalty or entropy bonus.
>
>
> - GRPO-KL=1e-4: We apply a small KL regularization term 1e-4 to control the divergence between the online policy and a reference policy. This helps prevent the model from drifting too far and encourages continued exploration.
>
>
> - DAPO: This is the DAPO baseline with decoupled clipping and dynamic sampling. No KL divergence penalty or entropy bonus.
>
>
> We observe that GRPO-KL=0 suffers from entropy collapse and the validation score stops improving after the entropy degrades to ~0 in ~500 steps. In contrast, DAPO and KL penalty both can mitigate entropy collapse as the validation scores continue to improve. Based on these findings, we adopt a combination of DAPO and a KL penalty in our final training setup.
>
>
> 3. *No analysis on the various model sizes.*
>
>
> We conduct further experiments by applying our training recipe to a larger model and different base model. We use Llama-3.1-Nemotron-Nano-8B-v1, which is a derivative based on llama-3.1-8B-Instruct, further finetuned on diverse reasoning tasks. Due to time constraints and compute limitations, we only scale training to ~1.4k steps. We provide detailed comparison with the base model across tasks as follows
> | Model                         | Math  | Code  | GPQA  | IFEval | Reasoning Gym |
> |-------------------------------|-------|-------|-------|--------|---------------|
> | Llama-3.1-Nemotron-Nano-8B-v1 | 68.26 | 61.20 | 52.76 |  66.88 |     12.15     |
> | ProRL-8B                      | 71.98 | 72.80 | 55.52 |  70.94 |     77.23     |
>
>
> Note: The above reported results on Code only consists of {apps, condecontests, codeforces, taco} and report the obtained continuous reward. All other benchmarks are the same as in the paper.
>
>
> 4. *Further scaled training results*
>
>
> We were able to further scale training for an additional 1k steps to a total 3k steps. We compare the new model with the original 2k model in our paper as follows:
> | Model                         | Math  | Code  | GPQA  | IFEval | Reasoning Gym |
> |-------------------------------|-------|-------|-------|--------|---------------|
> | ProRL-2k  | 60.14 | 37.49 | 41.78 |  66.02 |     59.06     |
> | ProRL-3k  | 61.69 | 41.03 | 41.32 |  70.85 |     62.49     |
>
>
> As shown above, prolonged RL continues to bring improved performance across most tasks, with only GPQA as an exception.
>
>
> 5. *Font sizes should be increased for the most of the figures*
>
>
> We apologize for the inconvenience and will take this into consideration when revising the final paper.
>
>
> 6. *Figure 1 (middle) demonstrates that base model produces some amount of outliers which score much higher in terms of creativity compared to samples after RL finetuning. What are those outliers? Are those just random garbage? And do you have any hypothesis why we don't see those for RL model?*
>
>
> Thanks for the great observation! The base model sometimes produces outliers with much higher creativity indices, likely because these tasks are highly out-of-distribution—unseen during the base model’s training and unfamiliar to it, such as the dice task in Reasoning Gym, which has near-zero pass@1 accuracy. After RL training, the model learns to handle these tasks (e.g., the dice task's pass@1 accuracy improves to 39.9%), making them more in-domain for the model.
>
>
> 7. *Is any weight decay used? AdamW is used as an optimizer, however no weight decay parameter is mentioned. I would also recommend to report all the hyperparameters in the form of a table in Appendix and not as parts of the text.*
>
>
> We use a weight decay of 0.01. We will include detailed hyperparameter configurations in the Appendix for our final version.

---

> > ### Comment · Reviewer_TrB6 · 2025-07-31
> >
> > I thank authors for detailed response and additional experiments. I'm raising my score.

---

### Decision · Program_Chairs · 2025-09-17

**Decision:**

Accept (poster)

**Comment:**

This paper challenges recent works that claim that RL fine-tuning does not improve the reasoning capabilities of the base model. They identify a number of issues with these prior works -- which essentially amounts to overfitting and early termination -- and propose a novel algorithm -- ProRL -- which mitigates these weaknesses with a KL regularization. There were concerns about whether the improved performance was due to improved reasoning or due to simply novel expose to the data, but through discussion and additional experiments, the authors successfully convinced the reviewers that the model fine-tuned with ProRL does indeed demonstrate enhanced reasoning over the base model. Additional ablations were asked for from the reviewers, and these were provided, e.g., hyper parameters, policy reset, and base model.

Given this is an active area of research (RL for fine-tuning / reasoning) and the quality of the paper, I advocate for acceptance as a poster.